# Nuclear crowding and nonlinear diffusion during interkinetic nuclear migration in the zebrafish retina

Afnan Azizi[1†], Anne Herrmann[2†], Yinan Wan[3], Salvador JRP Buse[1], Philipp J Keller[3], Raymond E Goldstein[2*], William A Harris[1*]

[1]Department of Physiology, Development and Neuroscience, University of Cambridge, Cambridge, United Kingdom; [2]Department of Applied Mathematics and Theoretical Physics, Centre for Mathematical Sciences, University of Cambridge, Cambridge, United Kingdom; [3]Howard Hughes Medical Institute, Janelia Research Campus, Ashburn, United States

**Abstract** An important question in early neural development is the origin of stochastic nuclear movement between apical and basal surfaces of neuroepithelia during interkinetic nuclear migration. Tracking of nuclear subpopulations has shown evidence of diffusion - mean squared displacements growing linearly in time - and suggested crowding from cell division at the apical surface drives basalward motion. Yet, this hypothesis has not yet been tested, and the forces involved not quantified. We employ long-term, rapid light-sheet and two-photon imaging of early zebrafish retinogenesis to track entire populations of nuclei within the tissue. The time-varying concentration profiles show clear evidence of crowding as nuclei reach close-packing and are quantitatively described by a nonlinear diffusion model. Considerations of nuclear motion constrained inside the enveloping cell membrane show that concentration-dependent stochastic forces inside cells, compatible in magnitude to those found in cytoskeletal transport, can explain the observed magnitude of the diffusion constant.

**\*For correspondence:**
R.E.Goldstein@damtp.cam.ac.uk (REG);
wah20@cam.ac.uk (WAH)

[†]These authors contributed equally to this work

## Introduction

The vertebrate nervous system arises from a pseudostratified epithelium within which elongated proliferating cells contact both the apical and basal surfaces. Within these cells, striking nuclear movements take place during the proliferative phase of neural development. More than 80 years ago, these movements, termed interkinetic nuclear migration (IKNM), were shown to occur in synchrony with their cell cycle (*Sauer, 1935*). Under normal conditions, nuclei of proliferating cells undergo mitosis (M) exclusively at the apical surface. During the first gap phase (G1) of the cell cycle, nuclei migrate away from this surface to reach more basal positions by synthesis phase (S), when DNA is replicated. In the second gap phase (G2), nuclei migrate rapidly toward the apical surface where they divide again (*Del Bene, 2011*; *Sauer, 1935*; *Baye and Link, 2007*; *Leung et al., 2011*; *Kosodo et al., 2011*; *Norden et al., 2009*). The molecular mechanisms that drive the rapid nuclear movement in G2 have been investigated in a number of tissues (*Norden, 2017*). In the mammalian cortex, they are thought to involve microtubules, as well as various microtubule motors and actomyosin (*Xie et al., 2007*; *Tsai et al., 2007*), while in the zebrafish retina, it appears to be the actomyosin complex alone that moves the nuclei to the apical surface during G2 (*Norden et al., 2009*; *Leung et al., 2011*). Nuclear movements during the majority of the cell cycle, in G1 and S phases, have been less thoroughly examined. Although similar molecular motors have been implicated (*Schenk et al., 2009*; *Tsai et al., 2010*), the underlying molecular processes remain unclear.

Importantly, IKNM is known to affect morphogenesis and cell differentiation in neural tissues (*Spear and Erickson, 2012*), as retinas with perturbed IKNM are known to develop prematurely and to display abnormalities in cell composition (*Del Bene et al., 2008*). Given this regulatory involvement of IKNM in retinal cell differentiation, a deeper understanding of the nuclear movements remains a major prerequisite for insights into the development of neural systems. On a phenomenological level, studies tracking individual nuclei in the zebrafish retina during the G1 and S phases have shown their movement to resemble a stochastic process (*Norden et al., 2009*; *Leung et al., 2011*), particularly in the form of the mean squared nuclear displacement versus time. When these relations are linear or slightly convex, they indicate a random walk (or persistent random walk), much as in ordinary thermal diffusion. During these periods, individual nuclei switch between apical and basal movements at random intervals, leading to great variability in the maximum basal position they reach during each cell cycle (*Baye and Link, 2007*). Similarly, in the mammalian cerebral cortex, the considerable internuclear variability in IKNM leads to nuclear positions scattered throughout the entire neuroepithelium in S phase (*Sidman et al., 1959*; *Kosodo et al., 2011*). In addition to the stochastic movements of nuclei during IKNM, there is also a slow basalward drift of the entire population of nuclei. As variable basalward-biased migration was observed in nuclear-sized microbeads inserted in between cells during IKNM in the mouse cortex (*Kosodo et al., 2011*), it seems likely that passive forces are involved in this drift. A number of possible explanations for these passive processes have been put forward. These suggestions include the possibility of direct energy transfer from rapidly moving G2 nuclei (*Norden et al., 2009*), as well as nuclear movements caused by apical crowding (*Kosodo et al., 2011*; *Okamoto et al., 2013*), that is an increase in nuclear packing density close to the apical tissue surface. Here, we present experiments and theoretical analyses to test both hypotheses, particularly that of apical crowding, and to assess quantitatively whether active forces are also necessary for basal drift.

While a linear scaling of the mean squared displacement with time is a hallmark of diffusive processes, there is now growing evidence in disparate systems of dynamics that exhibit such scaling, yet are decidedly different from conventional diffusion in other respects (*Wang et al., 2009*; *Leptos et al., 2009*). Thus, a full test of the apical crowding hypothesis requires the study of the entire spatio-temporal distribution of nuclei within the retinal tissue. Our work relies on the tracks of closely packed nuclei of zebrafish retinal progenitor cells (RPCs). The retina of the oviparous zebrafish is easily accessible to light microscopy throughout embryonic development (*Avanesov and Malicki, 2010*) and has been used for several studies of the movements of nuclei during IKNM (*Baye and Link, 2007*; *Del Bene et al., 2008*; *Norden et al., 2009*; *Sugiyama et al., 2009*; *Leung et al., 2011*). We find evidence for IKNM being driven by apical crowding and further develop this idea into a mathematical model. Given the seemingly stochastic nature of individual nuclear trajectories, we base the model on a comparison between IKNM and a simple diffusion process. The model reveals the remarkable and largely overlooked importance of simple physical constraints imposed by the overall tissue architecture and allows us to describe accurately the global distribution of nuclei as a function of time within the retinal tissue. In this way, we describe IKNM as a tissue-wide rather than a single-cell phenomenon. We further develop the model by examining the motion of nuclei within the constrained environment of the enveloping cell membrane. This allows for an estimate of the hydrodynamic drag experienced by the nuclei, and hence of their diffusivity if the system were in thermodynamic equilibrium. We conclude from the magnitude of the diffusivity extracted from the data that basalward migration of nuclei during IKNM cannot be due to thermal diffusion alone. Instead, the model indicates that a stochastic force comparable with that which could be generated by cytoskeletal transport mechanisms must drive nuclear movements during IKNM. Finally, we obtain a mathematical description of the stochastic trajectories of individual nuclei in the presence of a finite concentration of others. Simulations of these trajectories also confirm that IKNM can only be understood when taking interactions between individual nuclei into account and hint at the way in which nuclei interact in a tissue-wide fashion. This description raises new questions about how cells sense and respond to being crowded, and may shed light on other aspects of progenitor cell biology, such as the statistics of cell cycle exit and cellular fate choice.

## Results

### Generating image sets with high temporal resolution

We imaged fluorescently labeled nuclei of whole retinas of developing zebrafish at 2 min intervals, an optimal time period given the difficulty to track nuclei accurately over long times and the increased photobleaching with shorter intervals. We compared movies of retinas imaged at 2 min and at 20 s intervals over a period of 2 hrs and found that the improvement in temporal resolution made no difference to our analyses. This suggests that it is unlikely that there are important intervening movements that might complicate the analysis within each 2-min interval.

To follow the nuclei of all cells within a portion of the retina, we used H2B-GFP transgenic lines with GFP expression exclusively in the nuclei (*Figure 1A*). In order to achieve the desired temporal resolution without sacrificing image quality, fluorescence bleaching and sample drift must be minimized as much as possible. The retinas of H2B-GFP embryos were imaged using either a single-angle lightsheet microscope (see *Figure 1B* for a schematic) or an upright two-photon scanning microscope. Both of these methods yield images with minimal bleaching compared to other microscopic techniques (*Svoboda and Yasuda, 2006*; *Stelzer, 2015*). However, while the single-angle lightsheet can generate large stacks of images, it is very sensitive to lateral drift due to a small area of high resolution imaging. Therefore, some data sets were produced using two-photon microscopy, which, despite the limitations of scanning time, could produce areas of high-resolution images of sufficient size.

Both lightsheet and two-photon microscopes produced images of at least half the retina with a depth of at least 50 μm over several hours in 2-min intervals. The images were processed using a suite of algorithms (*Amat et al., 2015*) to compress them to a lossless format, Keller Lab Block (KLB), correct global and local drift, and normalize signal intensities for further processing. Automated segmentation and tracking, in three dimensions, of the nuclei were carried out through a previously published computational pipeline that takes advantage of watershed techniques and persistence-based clustering (PBC) agglomeration to create segments and Gaussian mixture models with Bayesian inference to generate tracks of nuclei through time (*Amat et al., 2014*). Two main parameters greatly affect tracking results, overall background threshold and PBC agglomeration threshold. To obtain best automated tracking results, ground truth tracks were created for a section of the retina over 120 min and were compared to tracks generated over a range of these two parameters. The best combination of the two parameters was chosen as the one with highest tracking fidelity and lowest amount of oversegmentation in that interval.

The most optimal combination of parameters yielded an average linkage accuracy, from each time point to the next, of approximately 65%. Hence, extensive manual curation and correction of tracks were required. Tracking by Gaussian mixture models (TGMM) software generates tracks that can be viewed and modified using the Massive Multi-view Tracker (MaMuT) plugin of the Fiji software (*Wolff et al., 2018*; *Schindelin et al., 2012*). A region of the retina with the best fluorescence signal was chosen and all tracks within that region were examined and any errors were corrected. The tracks consist of sequentially connected sets of 3D coordinates representing the centers of each nucleus (*Figure 1C*), with which their movement across the tissue can be mapped over time. For example, *Figure 1D* shows IKNM of a single nucleus tracked from its birth, at the apical surface of the retina, to its eventual division into two daughter cells.

### Analysis of nuclear tracks

This process yielded tracks for hundreds of nuclei, across various samples, over time intervals of at least 200 min. We used custom-written MATLAB scripts to analyze these tracks. The aggregated tracks of the main data set, in Cartesian coordinates, for all tracked lineages are shown in *Figure 2A*. Single tracks for any given time interval can be extracted and analyzed from this collection. In order to transform the Cartesian coordinates of the tracks into an apicobasal coordinate system, we drew contour curves at the apical surface of the retina (see *Figure 1A*) separating RPC nuclei from the elongated nuclei of the pigmented epithelium. We then calculated curves of best fit (second degree polynomials) in both the XY and YZ planes. Assuming that the apical cortex is perpendicular to the apicobasal axis of each cell, displacement vectors of the nuclei at each time point

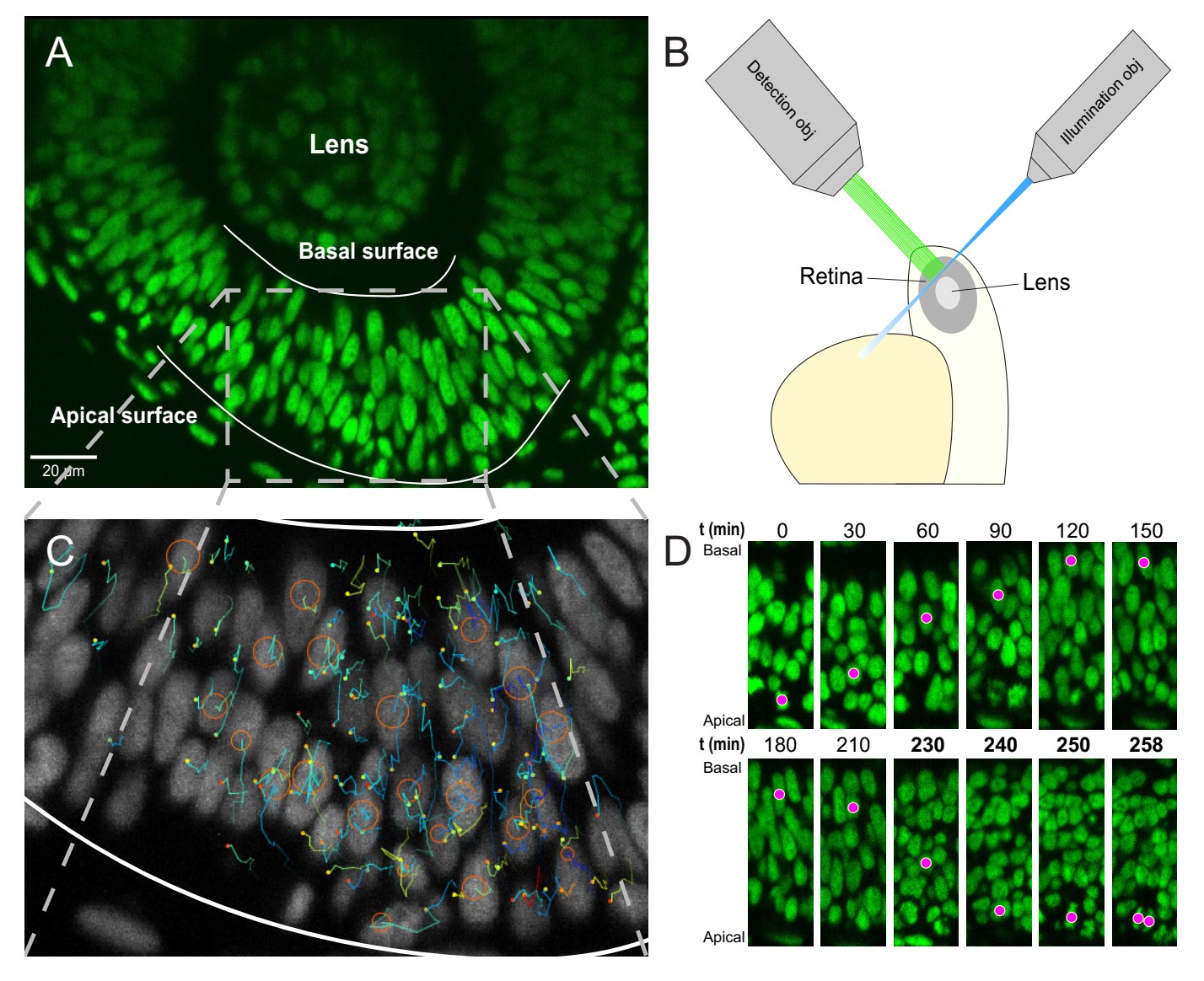

**Figure 1.** Imaging and tracking fluorescently labeled nuclei. (**A**) A transgenic H2B-GFP embryonic retina imaged using lightsheet microscopy at ~30 hpf. The lens, as well as apical and basal surfaces are indicated. (**B**) A schematic representation of single-angle lightsheet imaging of the retina. Laser light is focused into a sheet of light by the illumination objective and scans the retina. Fluorescent light is then collected by the perpendicular detection objective. (**C**) Track visualization and curation using the MaMuT plugin of Fiji. All tracks within a volume of the retina are curated and visualized. Circles and dots represent centers of nuclei, and lines show their immediate (10 previous steps) track. (**D**) The position of a single nucleus within the retinal tissue from its birth to its eventual division. The magenta dot indicates the nucleus tracked at various time points during its cell cycle. The last four panels are at shorter time intervals to highlight the rapid movement of the nucleus prior to mitosis.

can be separated into apicobasal and lateral components. Since, in IKNM, the apicobasal motion is that of interest, we used this component for our remaining analyses.

*Figure 2C,D* show the speed and position of tracked nuclei of the same data set, over the duration of their cell cycle, for all cells that went through a full cell cycle. While all nuclei behave similarly minutes after their birth (early G1) and before their division (G2), their speed of movement and displacement is highly variable for the majority of the time that they spend in the cell cycle (*Figure 2C, D*). Most daughter nuclei move away from the apical surface, within minutes of being born, with a clear basalward bias in their speed distribution (*Figure 2C*). This abrupt basal motion of newly divided nuclei has also been recently observed by others (*Leung et al., 2011*; *Shinoda et al., 2018*; *Barrasso et al., 2018*). However, immediately after this brief period, nuclear speeds become much

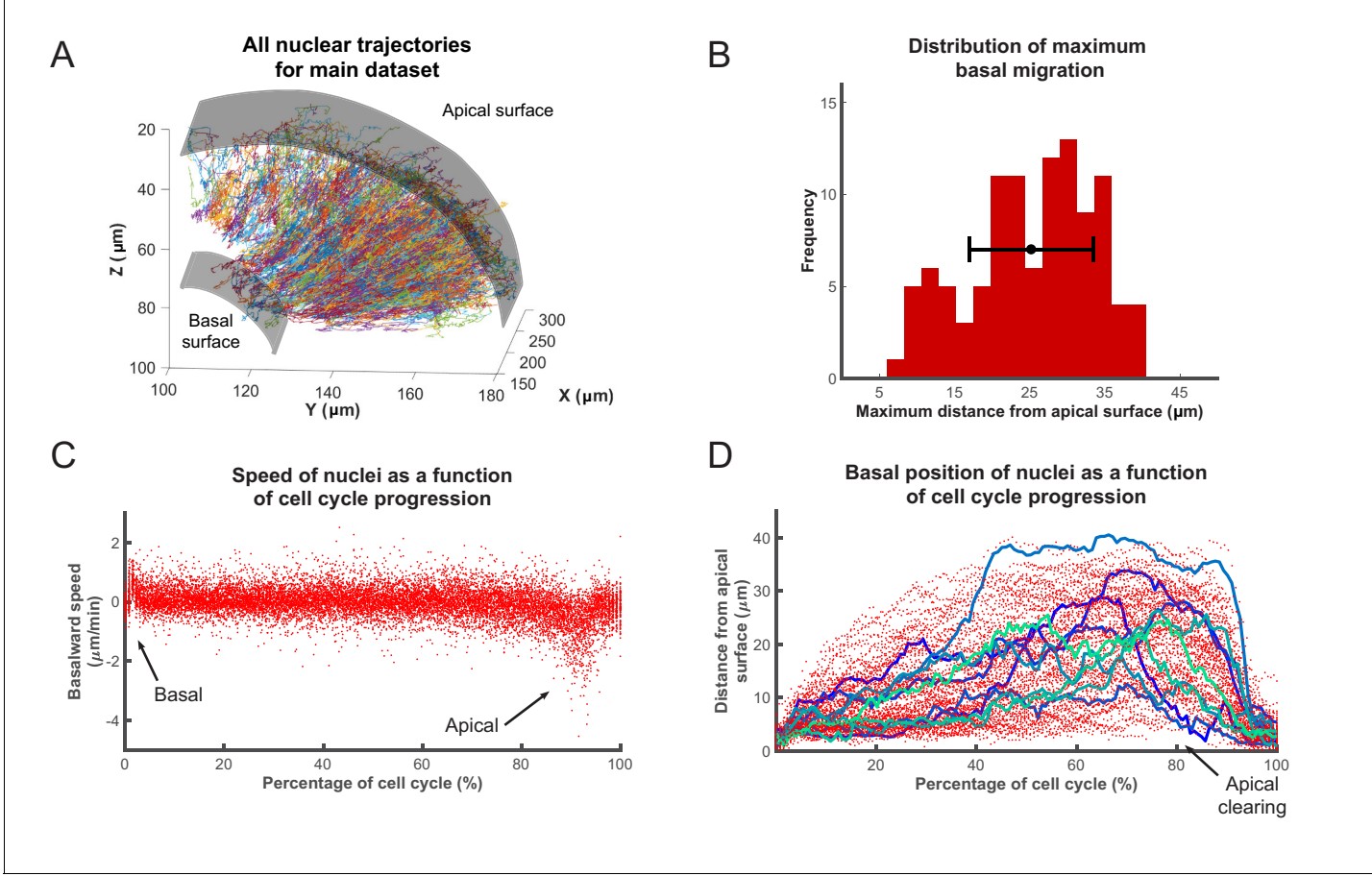

**Figure 2.** Analysis of nuclear tracks during IKNM. (**A**) Extracted trajectories of nuclei in three dimensions. All curated tracks of the main data set over 400 min in the region shown in *Figure 1C* are presented. (**B**) The distribution of maximum distances reached away from the apical surface by nuclei during their completed cell cycles. The mean and one standard deviation are shown. (**C**) The speed distribution of 106 nuclei over complete cell cycles. The cell cycle lengths of all nuclei were normalized and superimposed to highlight the early basal burst of speed, as well as pre-division apical rapid migration. The speeds between these two periods are normally distributed. (**D**) Position of the same nuclei as in (**C**) measured by their distance from the apical surface over normalized cell cycle time. Even though all nuclei start and end their cell cycle near the apical surface, they move out across the retina to take positions in all available spaces, creating an apical clearing as indicated. Tracks for 10 randomly chosen nuclei are shown as colored lines to highlight the variability in the traversed trajectories.

more equally distributed between basalward and apicalward, with a mean value near 0. Such a distribution is indicative of random, stochastic motion, which in turn leads to a large variability in the position of nuclei within the tissue (away from the apical surface) during the cell cycle (*Figure 2B*).

Interestingly, except during mitosis, we find an apical clearing of a few microns for dividing cells (*Figure 2D*). We checked to see if this was an artifact of measuring the distance to nuclear centers due to nuclear shape, as nuclei are rounded during M phase but are more elongated along the apicobasal axis at other times. We found no significant difference between the average length of the nuclear long axis when measured for 50 random nuclei right before their division (5.0 ± 0.7 μm) compared to 50 others chosen randomly from any other time point within the cell cycle (5.3 ± 1.1 μm), indicating that this clearing is likely to have a biological explanation, such as the preferential occupancy of M phase nuclei and surrounding cytoplasm at the apical surface during IKNM. We also performed the same measurements for 25 random nuclei 10 min after division when the average long axis length is significantly decreased by 0.8 fold (3.9 ± 0.5 μm). However, this measurement increased significantly in the following 10 min (4.8 ± 0.7 μm) to become similar to that at M phase.

## Basal movement of nuclei is driven like a diffusive process

Previous work has shown that when RPCs are pharmacologically inhibited from replicating their DNA, their nuclei neither enter G2 nor exhibit rapid persistent apical migration that normally occurs during the G2 phase of the cell cycle (*Leung et al., 2011*; *Kosodo et al., 2011*). A more surprising result of these experiments is that the stochastic movements of nuclei in G1 and S phases also slow down considerably during such treatment (*Leung et al., 2011*). It was, therefore, suspected that the migration of nuclei of cells in G2 toward the apical surface jostles those in other phases (*Norden et al., 2009*). We searched our tracks for evidence of such direct kinetic interactions among nuclei by correlating the speed and direction of movement of single nuclei with their nearest neighbors. These neighbors were chosen such that their centers fell within a cylindrical volume of a height and base diameter twice the length of long and short axes, respectively, of an average nucleus. *Figure 3A* shows the lack of correlation between the speed of movement of nuclei and the average speed of their neighbors. We further categorized the neighboring nuclei by their position in relation to the nucleus of interest (along the apicobasal axis), their direction of movement, and whether they were moving in the same direction of the nucleus of interest or not. None of the resulting eight categories of neighboring nuclei showed a correlation in their average speed with the speed of the nucleus of interest. Furthermore, we considered the movement of neighboring nuclei one time point (2 min) before or one time point after the movement of the nucleus of interest. Yet, we still found no correlation between these time-delayed and original speeds. These results suggest that there does not appear to be much transfer of kinetic energy between neighboring nuclei, and this is consistent with general considerations of the strongly overdamped character of motion at these length scales.

Another hypothesis advanced for the basal drift in IKNM is that the nuclear movements are driven by apical crowding (*Kosodo et al., 2011*; *Okamoto et al., 2013*). How apical crowding might result in basal IKNM can be understood by comparing IKNM to a diffusive process. In diffusion, a concentration gradient drives the average movement of particles from areas of high to areas of low concentration. However, despite the average movement being directed, each individual particle's trajectory is a random walk (*Reif, 1965*). Similarly, during IKNM a gradient in nuclear concentration is generated because nuclei divide exclusively at the apical surface. If basal IKNM were comparable to diffusion, this nuclear concentration gradient would be expected to result in a net movement of nuclei away from the area of high nuclear crowding at the apical side of the neuroepithelium (*Miyata et al., 2014*; *Okamoto et al., 2013*). Indeed, in IKNM each individual nucleus' trajectory resembles a random walk (*Norden et al., 2009*). Therefore, for the cells in the G1 and S phases (which account for more than 90% of the cell cycle time in our system), IKNM has, at least on a phenomenological level, the main features of a diffusive process.

To test further whether we can indeed describe IKNM using a model of diffusion, we first asked what would happen to the concentration gradient if we blocked the cell cycle in S phase, which inhibits both the apical movement of the nuclei in G2 and mitosis at the apical surface. If the

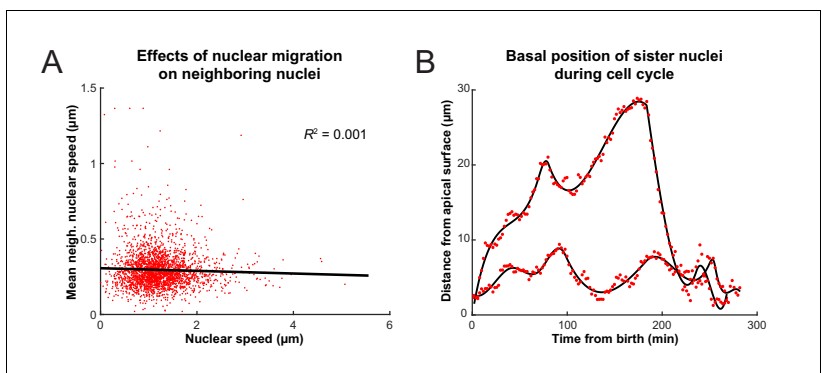

**Figure 3.** Interactions of nuclei in close proximity. (**A**) Average speed of nuclei neighboring a nucleus of interest as a function of the speed of that nucleus. (**B**) The positions of two sister nuclei at each time point imaged (red circles) over their complete cell cycle. The black lines are spline curves indicating the general trend of their movements.

comparison to diffusion were valid, we expect the blockage to abolish the build-up and maintenance of the concentration gradient. We, therefore, compared the normally evolving distribution of nuclei in a control retina with that measured from a retina where the cell cycle was arrested at S phase using a combination of hydroxyurea (HU) and aphidicolin (AC) (*Leung et al., 2011*; *Icha et al., 2016*). These compounds inhibit DNA polymerase and ribonuclear reductase, respectively, to halt DNA replication (*Baranovskiy et al., 2014*; *Singh and Xu, 2016*). In the HU-AC-treated retina, we counted the number of nuclei in a three-dimensional section of the tissue containing approximately 100 nuclei, at equal time intervals, starting with 120 min after drug treatment. The delay ensured that almost all cell divisions, from nuclei that had already completed the S phase at the time of treatment, had taken place. These results are shown in *Figure 4A,C*, in which retinal tissue is approximated as a spherical shell of apical radius $a$ and the rescaled coordinate $\xi = r/a$, where $r$ is the distance from the center of the lens, is presented on the x-axis. As expected from the diffusion model (*Figure 4D*), over the course of 160 min, the mean of the nuclear distribution moved further toward the basal surface in treated retinas, and the concentration difference between the apical and basal surfaces diminished (*Figure 4B,C*). In contrast, in control retinas the mean of the nuclear distribution moved toward the apical surface (*Figure 4A,C*) as the gradient continued to build up. Hence, these results support the suitability of a diffusive model to describe the basal nuclear migration during IKNM.

## An analytical diffusion model of IKNM

To investigate whether a diffusion model provides a quantitative description of IKNM, we focused on the crowding of nuclei at the apical side of the tissue. In mathematical terms, crowding creates a gradient in nuclear concentration $c$ along the apicobasal direction of the retina. If we assume there is no dependence of the nuclear concentration on the lateral position within the tissue then we require a diffusion equation for the nuclear concentration $c(r, t)$ as a function only of the apicobasal distance $r$ and time $t$. The retina can be approximated as one half of a spherical shell around the lens, and thus we use spherical polar coordinates with the origin of the coordinate system at the center of the lens, the basal surface at $r = b$ and the apical surface at $r = a$ (*Figure 5B*). We first consider the simplest diffusion equation for this system, in which there is a diffusion constant $D$ independent of position, time, and $c$ itself, namely

$$\frac{\partial c}{\partial t} = \frac{D}{r^2}\frac{\partial}{\partial r}\left(r^2\frac{\partial c}{\partial r}\right). \tag{1}$$

We seek to determine $D$ from the experimental data of the concentration profile $c(r, t)$. Note that in this parsimonious view of modeling we have not included a 'drift' term of the kind that is expected to be present at the very late stages of IKNM, when nuclei return to the apical side.

In addition to *Equation 1*, we must specify the boundary conditions appropriate to IKNM. Since nuclei only divide close to the apical surface of the tissue, we treat mitosis as creating an effective influx of nuclei through the apical boundary. To quantify this influx, we extracted the number of cells $N(t)$ as a function of time. As during the stages of development examined here cells are neither dying nor exiting the cell cycle (*Biehlmaier et al., 2001*), we assumed that the number of cell divisions is always proportional to the number of currently existing cells. This assumption predicts an exponential increase in the number of cells or nuclei, over time, as was recently confirmed by *Matejčić et al., 2018*:

$$N(t) = N_0 e^{t/\tau}, \tag{2}$$

where $N_0$ is the initial number of nuclei and $\tau = T_P/\ln 2$, with $T_P$ the average cell cycle length. *Figure 5A* shows the agreement between the theoretically predicted curve $N(t)$ with the experimentally obtained numbers of nuclei over time. Having obtained $N_0$ and $T_P$ from our experimental data, the predicted curve has no remaining free parameters and thus no fitting is necessary. Using *Equation 2*, we formulate the influx boundary condition as

$$D\frac{\partial c}{\partial r}\bigg|_{r=a} = \frac{1}{S}\frac{\partial N(t)}{\partial t} = \frac{N_0}{S\tau}e^{t/\tau}, \tag{3}$$

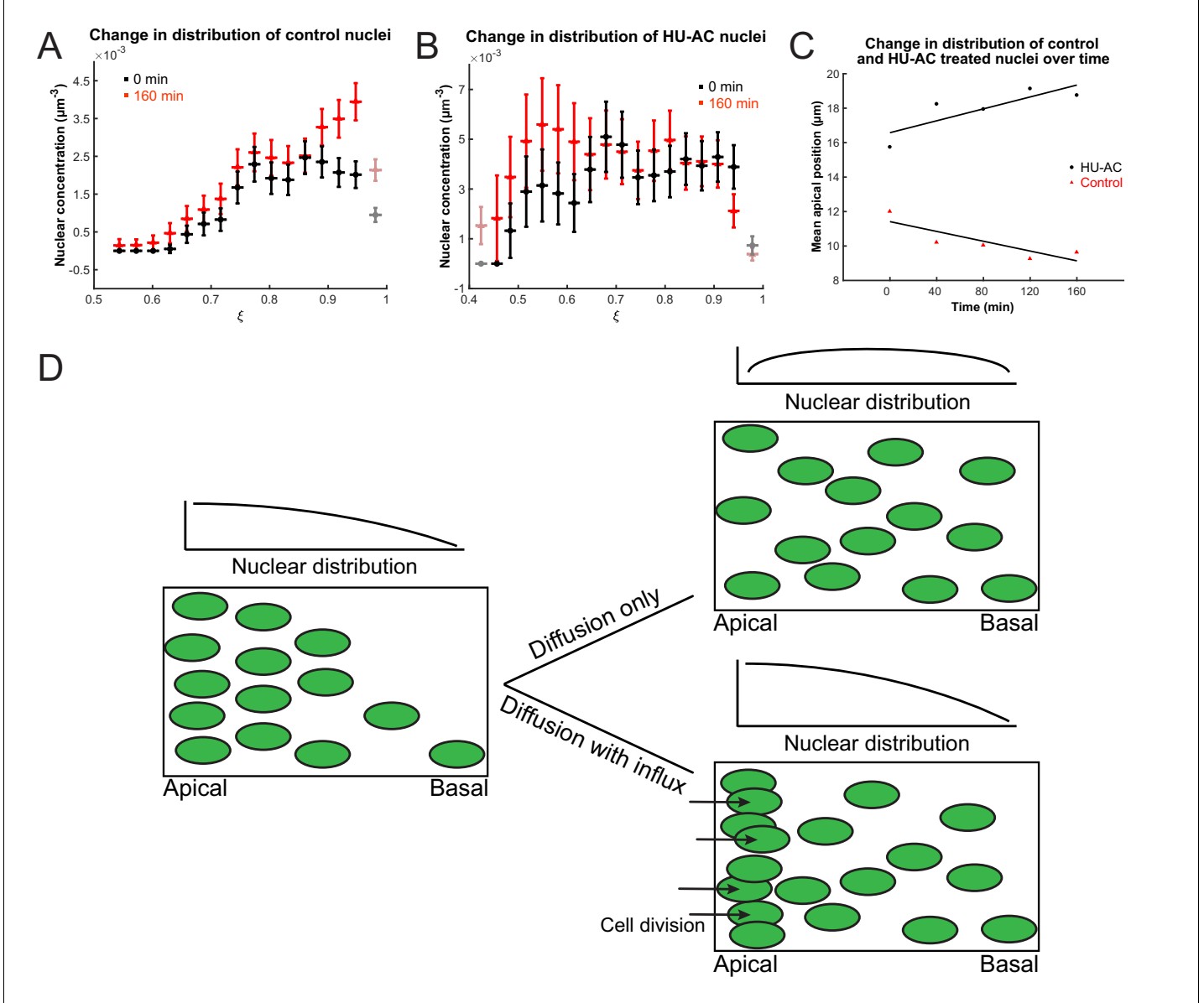

**Figure 4.** Nuclear concentration gradient across the apicobasal axis of the retina. The concentration of nuclei is higher near the apical surface compared to the basal surface. (**A**) In the control retina, the nuclear concentration gradient builds up over time. (**B**) Blocking apical migration and division of nuclei, by inhibiting S phase progression, leads to a shift in the distribution of nuclei toward the basal surface in the HU-AC treated retina. In A and B, the coordinate $\xi = r/a$ is used, where $a$ is the radius of the apical surface and $r$ the distance from the center of the lens. (**C**) The shift in the distribution of nuclei under HU-AC treatment when compared to the untreated retina. The average distance of nuclei away from the apical surface increases consistently over time in the absence of cell division, but remains the same when new nuclei are constantly added at the apical surface. (**D**) A schematic of how a diffusion model would work in the context of IKNM in the retina. A concentration gradient of nuclei (left) would drive the net movement of nuclei from the apical surface to the basal surface. However, without maintenance of the gradient, the drive for this net migration is lost (top right). In the retina, the gradient is maintained through cell divisions at the apical surface, modeled as a one-way influx across the apical surface (bottom right), continuously driving the net movement basally.

with $S$ the apical surface area of our domain of interest. In contrast to the apical side of the tissue, there is no creation (or depletion) of nuclei at the basal side (*Matejčić et al., 2018*), and hence a no-flux boundary condition,

$$\left.\frac{\partial c}{\partial r}\right|_{r=b} = 0. \qquad (4)$$

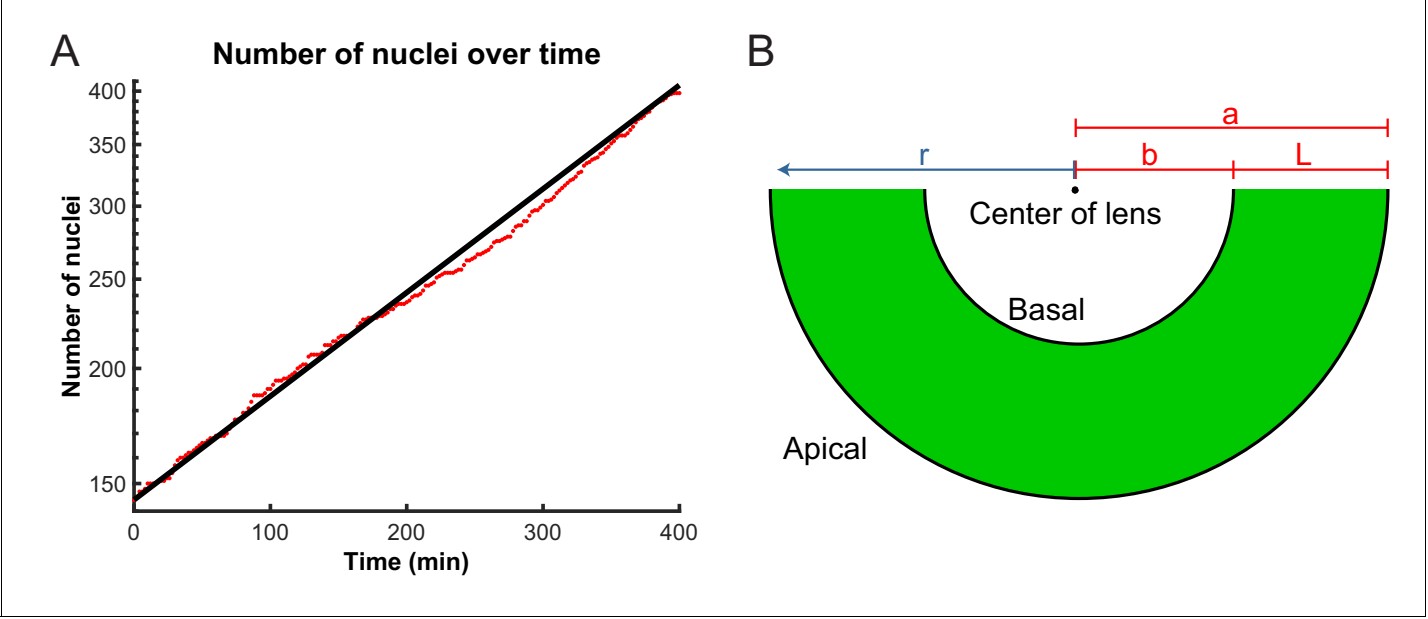

**Figure 5.** Model parameters extracted from experimental data. (A) Number of nuclei grows exponentially during the proliferative stage of the retinal development. A line can be fitted to the log-lin graph of nuclear numbers as a function of time to extract the doubling time (cell cycle length) in this period. (B) A schematic of the retina indicating the variables used in the diffusion model of IKNM. a: distance from center of lens to apical surface; b: distance from center of lens to basal surface; L: thickness of the retina; r: distance from center of lens for each particle.

The position $r = b$ where this basal boundary condition is applied could change throughout tissue development. *Matejčić et al., 2018* found that a basal exclusion zone, where nuclei cannot enter due to accumulation of basal actin, exists in the zebrafish retina before approximately 42 hpf. Before this point in development, the no-flux boundary condition is applied at the tissue radius where the nuclear exclusion zone begins, while later in development, the no-flux boundary condition should be applied at the position of the actual basal cell surfaces. Here, we only model early stages of embryonic development well before the disappearance of the basal exclusion zone, therefore the location $r = b$, where we apply our basal boundary condition, is chosen such that we only consider the region of the retinal tissue actually accessible to moving nuclei during IKNM. Thus, taken together, *Equations 1, 3 and 4* fully specify this simplest mathematical model of IKNM.

In solving these equations to find the concentration of nuclei $c(r,t)$ in the retinal tissue it is convenient to introduce dimensionless variables for space and time,

$$\xi = \frac{r}{a}, \qquad s = \frac{Dt}{a^2}, \tag{5}$$

and further define the purely geometric parameter $\rho = b/a < 1$. The exact solution for the nuclear concentration, whose detailed derivation is given in the Appendix, is

$$c(\xi,s) = \sum_{i=1}^{\infty} \left( h_i e^{-\lambda_i^2 s} + \frac{\alpha_i f_0}{\sigma + \lambda_i^2} e^{\sigma s} \right) H_i(\xi) + \frac{1}{1-\rho} \left( \frac{1}{2} \xi^2 - \rho \xi + g_0 \right) f_0 e^{\sigma s}. \tag{6}$$

The first terms within parentheses describe the decay over time of the initial condition $c(\xi, s = 0)$. Here, $\lambda_i$ are the eigenvalues and $H_i(\xi)$ the eigenfunctions of the radial diffusion problem, and the coefficients $h_i$ are determined from the experimental initial conditions (see Materials and methods). The second terms within the sum and the final term on the right hand side of *Equation 6* are constructed such that the solution fulfills the boundary conditions *Equation 3* and *Equation 4*. In the last term, the constant $g_0$ was obtained using the constraint that the volume integral of the initial concentration yields the initial number of nuclei $N_0$. $f_0$, $\sigma$ and $\alpha_i$ emerge within the calculation of the solution and are specified in the Appendix. Thus, the diffusion constant $D$ in *Equations 1 and 6* is the only unknown.

## The linear model is accurate at early times

To determine the effective diffusion constant $D$ from the data, the experimental distribution of nuclei in the retinal tissue was first converted into a concentration profile. Then, the optimal $D$-value, henceforth termed $D^*$, was obtained using a minimal-$\chi^2$ approach. The value obtained within the linear model for a binning width of 3 μm and an apical exclusion width of 4 μm is $D_{\mathrm{lin}}^* = 0.17 \pm 0.07$ μm²/min. Using this, we can examine the decay times of the different modes in the first term of *Equation 6*. The slowest decaying modes are the ones with the smallest eigenvalues $\lambda_i$ and we find that the longest three decay times are $\mathcal{T}_1 \approx 1325$ min, $\mathcal{T}_2 \approx 350$ min and $\mathcal{T}_3 \approx 158$ min. This shows that indeed all three terms of *Equation 6* are relevant on the timescale of our experiment and need to be taken into account when calculating the concentration profile. The corresponding plots of $c(\xi, s)$ are shown in *Figure 6A–C*. As can be seen from this figure, the diffusion model fits the data very well at early times, $t \leq 200$ min after the start of the experiment at 24 hpf (see Materials and methods). However, for $t \geq 200$ min the model does not fit the data as well; the

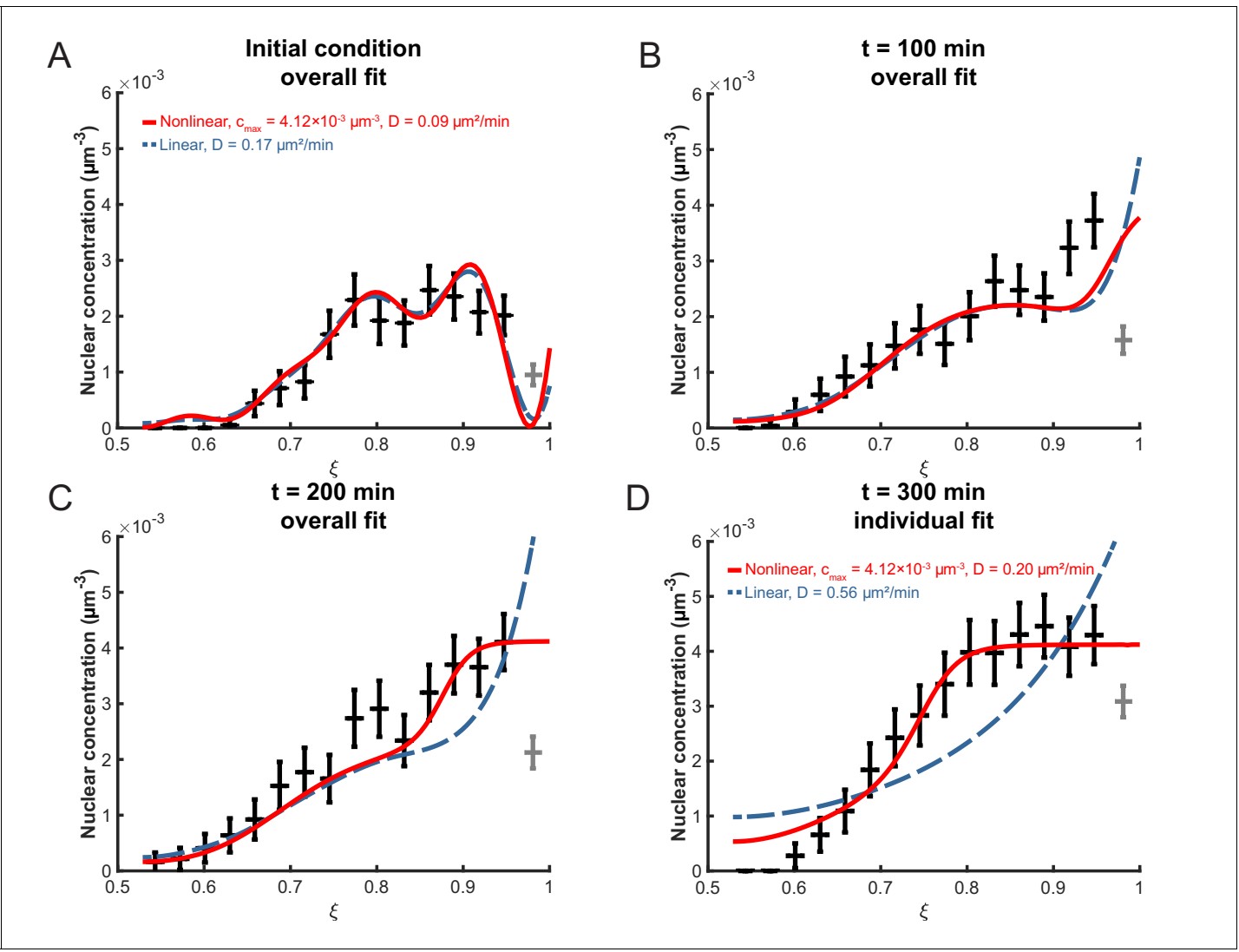

**Figure 6.** Fitting the linear and nonlinear models to the distribution of nuclei over time. (A) The initial experimental concentration profile of nuclei at $t = 0$ min, as well as the calculated initial condition curves (see Materials and methods *Equation 17*) for the linear (red solid line) and nonlinear (blue dashed line) models. The fit of the models to experimental distribution of nuclei after 100 min (B), 200 min (C), and 300 min (D) are shown. For the first three graphs, the best fits over all 100 intervening time points were used with the corresponding diffusion constants shown in (A). For t = 300 min, the best fits at that time point only were used with the corresponding diffusion constants indicated.

experimentally observed nuclear concentration levels off at a value between $4.00 \times 10^{-3}$ $\mu m^{-3}$ and $4.50 \times 10^{-3}$ $\mu m^{-3}$ (*Figure 6D*), an aspect that is not captured by this model of linear diffusion.

One particular aspect of the biology that the linear model neglects is the spatial extent of the nuclei. In the linear diffusion model, particles are treated as point-like and non-interacting. However, our microscopy images (see *Figure 1A*) clearly indicate that the nuclei have finite incompressible volumes, so that their dense arrangement within the retinal tissue would lead to steric interactions once the nuclear concentration is sufficiently high. Moreover, the packing density of nuclei can not exceed a maximum value dictated by their geometry. Therefore, we next examined whether accounting for such volume and packing density effects leads to a more accurate theory describing the nuclear distribution during IKNM.

## Nonlinear extension to the model

When the diffusion *Equation 1* is written in the following form

$$\frac{\partial c}{\partial t} = D \frac{1}{r^2} \frac{\partial}{\partial r} \left\{ r^2 c \frac{\partial}{\partial r} \left[ \frac{\partial}{\partial c}(c \ln c) \right] \right\}, \tag{7}$$

we can identify the term $c \ln c$ as proportional to the entropy density $\mathscr{S}$ of an ideal gas, and its derivative with respect to $c$ as a chemical potential. In an ideal gas, all particles are treated as point-like and without mutual interactions. In order to include the spatial extent of particles (i.e. the spatial extent of nuclei in this case), we must estimate the entropy in a way that accounts for the maximum concentration allowable given steric interactions. This is a well-studied problem in equilibrium statistical physics, in which, purely as a calculation tool, it is useful to consider space as divided up into a lattice of sites. Each of these sites can be either empty or occupied by a single particle. In this 'lattice gas' model, the discrete sites assure a minimum distance of approach for particles and thus effectively introduce a particle size and, correspondingly, a maximum particle concentration $c_{max}$ (*Huang, 1987*). In this system, a useful approximation to the entropy is

$$\mathscr{S}_{\text{latticegas}} \propto c \ln c + (c_{max} - c) \ln(c_{max} - c). \tag{8}$$

Substituting this expression for the term $c \ln c$ in *Equation 7*, we obtain the nonlinear diffusion equation

$$\frac{\partial c}{\partial t} = D \frac{1}{r^2} \frac{\partial}{\partial r} \left( r^2 \frac{c_{max}}{c_{max} - c} \frac{\partial c}{\partial r} \right). \tag{9}$$

The term 'nonlinear' refers to the mathematical structure of the newly obtained *Equation 9*. In the mathematical classification, an equation is linear in a certain variable if this variable only appears raised to the power one within the equation. For example, the simplest diffusion *Equation 1* is linear in $c$ and all its derivatives with respect to $r$ and $t$, such as $\partial c / \partial t$. In contrast, in *Equation 9* the term $c_{max}/(c_{max} - c)$ appears which is proportional to $c^{-1}$. Hence, *Equation 9* is said to be nonlinear. The additional nonlinear term in *Equation 9* (as compared to *Equation 1*) is an important aspect of the model as it arose from the introduction of the spatial extent of the nuclei and their maximum possible packing density $c_{max}$. This effect also has to be taken into account in the boundary conditions. Adjusting the boundary conditions at the apical side accordingly leads to

$$D \frac{c_{max}}{c_{max} - c} \frac{\partial c}{\partial r} \bigg|_{r=a} = \frac{N_0}{S\tau} e^{t/\tau}, \tag{10}$$

while the basal boundary condition remains the same as *Equation 4*. Together, *Equation 9* and the boundary conditions in *Equations 4 and 10* represent an extension to the diffusion model for IKNM, which now accounts for steric interactions between the nuclei. The maximum concentration $c_{max}$ incorporated in this model was obtained, as described in the Materials and methods, by considering a range of nuclear radii and the maximum possible packing density for aligned ellipsoids (*Donev et al., 2004*).

Similar to fitting the linear model, we also need to establish a description of the initial condition. To make both models consistent with each other, we employ the linear model's initial condition, *Equation 6* at $s = 0$ with $h_i$ as obtained from *Equation 17* (*Figure 6A*). The concentration profile in

the nonlinear model and its derivative were obtained numerically using the MATLAB pdepe solver. Fitting this concentration profile to the data was by means of a minimal-$\chi^2$ approach as well. When the optimization takes data points up to $t = 200$ min into account, we find $D^*_{\mathrm{nonlin}} = 0.09 \pm 0.05$ μm²/min (*Figure 6*, *Table 1*). As can be seen, by choosing $c_{\max}$ correctly, an excellent fit to the data can be obtained, particularly to the flattened part of the distribution at later times near the apical side ($\xi \sim 1$), where the linear model fails. These results show that a lattice-gas based diffusion model is indeed suitable to describe time evolution of the nuclear concentration profile of the zebrafish retina during IKNM over several hours of early development.

## Basalward IKNM is not due to thermal diffusion but is compatible with cytoskeletal transport

This diffusion model, with the calculated diffusion constant $D^*_{\mathrm{nonlin}} = 0.09 \pm 0.05$ μm²/min obtained from the nonlinear implementation, allows us to probe the physical and biological considerations that could set its scale. Notably, at low nuclear densities, $c \ll c_{\max}$, the term $c_{\max}/(c_{\max} - c)$ in *Equation 9* tends to unity, and the ordinary diffusion *Equation 1* with $D^*_{\mathrm{lin}} = D^*_{\mathrm{nonlin}}$ is recovered. We can thus make use of its well-known properties for further evaluation. First, we assess whether nuclei in IKNM move due to free equilibrium thermal diffusion in a fluid. If so, the diffusion constant obeys the Stokes-Einstein equation (*Einstein, 1905*)

$$D_{\mathrm{thermal}} = \frac{k_{\mathrm{B}}T}{\zeta}, \tag{11}$$

where $k_{\mathrm{B}} = 1.38 \times 10^{-23}$ JK⁻¹ is the Boltzmann constant, $T$ is the absolute temperature, and $\zeta$ is the drag coefficient for the particle, the constant of proportionality between the speed with which it moves and the force applied. For a spherical particle of radius $\Re$ in a fluid of viscosity $\eta$, the reference value is $\zeta_0 = 6\pi\eta\Re$. If we assume that the particles move in water at 25 °C, for which $\eta \approx 9 \times 10^{-4}$ Pas, and if we approximate the nuclei as spheres with $\Re = 3.5$ μm, corresponding to the maximum nuclear concentration $c_{\max} = 4.12 \times 10^{-3}$ μm⁻³ (as in *Figure 6*), we obtain $D_{\mathrm{thermal}} \approx 4.2$ μm²/min. This value is about 50 times *larger* than the measured value of $D^*_{\mathrm{nonlin}}$, implying that freely diffusing nuclei in water would be vastly more mobile than seen during IKNM.

While the free thermal diffusivity of nuclei serves as a useful reference quantity, nuclei clearly do not move in pure water, nor in an unbounded fluid. The viscosity of the cytoplasm is likely much higher than that of water due to the high number of organelles and polymeric components present; a higher viscosity leads to a lower diffusion constant via the Stokes-Einstein relation (*Equation 11*). Similarly, the slender shape of the individual cells within pseudostratified epithelia (*Norden, 2017*) would imply that a considerable amount of energy is required to transport fluid through the narrow region between the nucleus and the membrane.

In order to understand the effects of membrane confinement on fluid transport, it is useful to consider a minimal energetic description of the cell shape. That is provided by an energy $\mathcal{E}$ that incorporates membrane elasticity, through a bending modulus $\kappa$, and surface tension $\gamma$,

$$\mathcal{E} = \int dS \left\{ \frac{\kappa}{2}\mathcal{H}^2 + \gamma \right\}, \tag{12}$$

where $dS$ is the element of surface area and $\mathcal{H}$ is the mean curvature. For a cylindrically symmetric shape given by a function $\delta(z)$, $dS = 2\pi\delta\sqrt{1 + \delta_z^2}$ and

**Table 1.** List of best-fit diffusion constants $D^*$, their standard deviations and probabilities for the studied conditions.

| | $D^*_{\mathrm{nonlin}}$ (μm²/min) | $\sigma_D$ (μm²/min) | $P_\chi(\chi^2; \nu)$ |
|---|---|---|---|
| Normal | 0.09 | 0.05 | 0.49–0.51 |
| Normal (repeat sample) | 0.10 | 0.06 | 0.47–0.48 |
| High T | 0.13 | 0.08 | 0.42 |
| Low T | 0.06 | 0.05 | 0.69–0.7 |

$$\mathcal{H} = \frac{\delta_{zz}}{\left(1 + \delta_z^2\right)^{3/2}} - \frac{1}{\delta\sqrt{1 + \delta_z^2}}, \qquad (13)$$

where $\delta_z$ stands for $d\delta/dz$, etc. The equilibrium shape of a membrane is that which minimizes *Equation 13* subject to constraints such as boundary conditions and/or a given enclosed volume.

As first understood in the context of the so-called 'pearling instability' of membranes under externally imposed tension (*Bar-Ziv and Moses, 1994*; *Nelson et al., 1995*; *Goldstein et al., 1996*), narrow necks emerge as characteristic equilibrium structures when the dimensionless ratio $\gamma R_\infty^2/\kappa$ is much larger than unity, where $R_\infty$ is a characteristic tube radius imposed far from the neck (e.g. the nuclear radius $\mathfrak{R}$). In this limit, the neck radius is on the order of $\sqrt{\kappa/\gamma}$. For fluid membranes, it is known that $\kappa \sim 20 k_\mathrm{B} T$ (*Helfrich, 1973*), while the magnitude of tension (an energy per unit area) is such that the surface energy associated with a molecular area is comparable to thermal energy; $\gamma \ell^2/k_\mathrm{B} T \sim 1$, where $\ell$ is a molecular dimension (e.g. 1 nm). Thus, $\gamma$ may be as large as $\sim 10^{-5}$ $\mathrm{Jm}^{-2}$ and $\gamma \mathfrak{R}^2/\kappa$ is very large indeed ($\sim 10^5$).

To illustrate the kinds of shapes that are energetic minima of *Equation 12*, we show in *Figure 7* that which arises when we impose (i) an overall aspect ratio of $\sim 20$ for the cell, as measured by *Matejčić et al., 2018*, (ii) cell radii of 1.98 and 0.94 μm at the apical and basal sides of the tissue, respectively, as determined from that aspect ratio and the approximate length $L$ of cells in our experiment, and (iii) position of the nucleus at the midpoint of the cell, with a radius $\mathfrak{R} = 3.5$ μm. The details of calculations are given in the Appendix. As the necks become extremely narrow in the relevant limit, we have taken a smaller value of $\gamma$ to illustrate the basic effect. Because the gap between the membrane and the enveloped sphere is so thin, we have set the membrane radius equal to that of a sphere with membrane radius $\mathfrak{R}_\mathrm{tube}$ over some angular extent and minimized the energy with respect to the position of the last contact point, as detailed in the Appendix.

The similarity of this shape to those described in the literature suggests that this model is a useful starting point for the discussion of the fluid dynamics of nuclear motion during IKNM. Recently, *Daniels, 2019* considered the transport of a sphere through the fluid contained within a cylindrically symmetric tight-fitting tubular membrane with bending modulus $\kappa$ and surface tension $\gamma$, much like the geometry of cells undergoing IKNM. At a finite temperature $T$, the membrane will exhibit thermally driven shape fluctuations which, as shown by *Helfrich, 1978*, produce a repulsive interaction with the nearby sphere, swelling the gap. In the limit of large tension (appropriate for a tight-fitting membrane), the calculation simplifies to yield the result

$$\zeta_\mathrm{tube} = 32 \zeta_0 \left( \frac{\kappa}{k_\mathrm{B} T} \frac{\gamma \mathfrak{R}^2}{k_\mathrm{B} T} \right)^{2/3}, \qquad (14)$$

where, for ease of interpretation, we have written the factors within parentheses as a product of two convenient dimensionless ratios. As the nuclear radius is micron-sized, we find $\gamma \mathfrak{R}^2/k_\mathrm{B} T \sim 10^7$, which in turn implies a drag coefficient ratio on the order of $10^5$ and diffusivities $D_\mathrm{tube} \approx (1-5) \times 10^{-6} D_\mathrm{thermal}$. Because of the very close spacing between the membrane and nucleus and the high viscous drag associated with such a geometry, these values are about 3 to 4 orders of magnitude smaller than the measured $D_\mathrm{nonlin}^*$. This is without considering changes in the cytoplasmic viscosity, which would decrease the value of $D$ even further. Therefore, we conclude that the nuclear movements in IKNM cannot be due to thermal diffusion, but must be actively driven, for example through cytoskeletal transport.

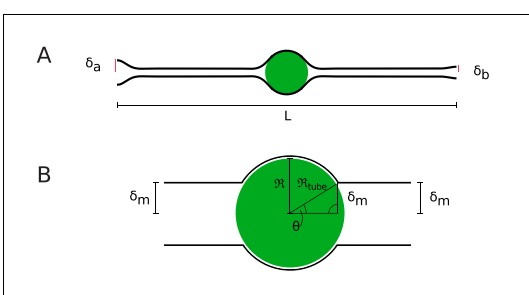

**Figure 7.** Cell shapes. (A) Equilibrium cell shape obtained from minimization of elastic energy, with specified radii $\delta_a = 1.98$ μm and $\delta_b = 0.94$ μm at apical and basal sides. Here, the length $L$ of the cell is taken to be 55 μm. (B) Coordinate system defined in *Daniels, 2019*, where $\mathfrak{R}$ is the nuclear radius and $\mathfrak{R}_\mathrm{tube}$ and $\theta$ are the radius of the membrane tube around the nucleus and the opening angle of the membrane, respectively.

We can turn to a more microscopic interpretation of the value of the diffusion constant. At low nuclear concentrations, when *Equation 1* holds, the behavior of individual particles can be described using the overdamped Langevin equation (compare to *Lemons and Gythiel, 1997*)

$$\zeta \frac{\partial r}{\partial t} = \mathscr{F}(t) \tag{15}$$

where $\mathscr{F}(t)$ is a stochastic force. In the standard way, if we average over realizations of the random force $\mathscr{F}(t)$ and integrate in time, the mean squared displacement $<r(t)^2> = \Gamma t/\zeta^2$ is obtained, where $\Gamma = \int dq Q(q)$, with $Q = <\mathscr{F}(t')\mathscr{F}(t'')>$ the correlation function of the stochastic force between time points $t'$ and $t''$ and $q = t' - t''$. For systems at densities low enough for *Equation 1* to hold, we know further that $<r(t)^2> = 6Dt$, leading to the result

$$\Gamma = 6\zeta^2 D, \tag{16}$$

expressing the unknown quantity $\Gamma$ in terms of the measured diffusion constant and the friction coefficient. Using the numerical values quoted above, we find $\Gamma \approx (1.2 \times 10^{-18} - 3.4 \times 10^{-17})$ N$^2$s. As the units of $\Gamma$ are force$^2 \times$ time, we can estimate the underlying forces if we know their correlation time. As most molecular processes of cytoskeletal components have characteristic time scales of 10 ms to 1 s, we obtain forces in the range of 1–50 nN. This result is compatible with cytoskeletal transport under the assumption that the nucleus is transported either by multiple molecular motors at once, since each molecular motor protein typically exerts forces on the order of several pN, or through typical forces arising from polymerization of cytoskeletal components, which are in the same range (*Peskin et al., 1993*; *Footer et al., 2007*).

## A stochastic model for the movement of individual nuclei reveals a potential microscopic mechanism for concentration-dependent IKNM

Having obtained an interpretation of the diffusion constant $D^*$ as arising from cytoskeletal transport throughout the cell cycle, and not only during the apicalward movement of the nuclei during G2, we turn to an interpretation of the concentration dependence of IKNM that results from nuclear crowding (*Equation 9*). To this end, we seek an extension to the stochastic dynamics of individual nuclei (*Equation 15*) that corresponds to the concentration evolution in the nonlinear diffusion *Equation 9*. In general, there are two different ways to achieve such a correspondence. In the first, an additional force $F_{\text{external}}$ is introduced into the Langevin *Equation 15*, which describes the average effect of surrounding nuclei on the individual nucleus in question and is thus concentration-dependent. In the second, we make direct use of the fact that $D_{\text{nonlin}} c_{\max}/(c_{\max} - c) \to D_{\text{lin}}$ as $c \to 0$. Inverting this relationship and applying it to the expression $\Gamma = 6\gamma^2 D$ for the low concentration case, we can also extend the Langevin *Equation 15* by making $\Gamma$ concentration-dependent, that is, $\Gamma = 6\gamma^2 D^*_{\text{nonlin}} c_{\max}/(c_{\max} - c)$.

Using both models, we can simulate individual nuclei in the experimental environment they experience during IKNM, namely the time-varying nuclear distribution across the retinal tissue that we found as the solution of the nonlinear model. Simulating several nuclei where each single one corresponds exactly to one nucleus in the experiment gives us a means to replicate the processes that took place in the tissue over a larger period of time. From such a simulation, we can also extract a mean squared displacement curve (MSD curve) that corresponds to the MSD curve calculated from the experimental nuclear trajectories. Of course, because our simulations are based on a stochastic equation, suitable averaging over realizations of the stochastic force are used to obtain statistically significant results.

*Figure 8* shows the range of possible MSD curves for simulations of the low concentration model described by *Equation 15* and those with the two possible high-concentration extensions, each represented by a shaded area. Shown also is the experimental MSD curve obtained from the very same nuclei used in the numerics. As can be seen, the experimental curve only agrees with the model that assumes a concentration-dependent value of $\Gamma$, and not the low-concentration model from *Equation 15*. In addition, the experimental curve does not agree with the possibility of including the effects of surrounding nuclei as an independent, additional force. These results have two implications. First, they lend further support to the notion raised above that IKNM cannot be understood as

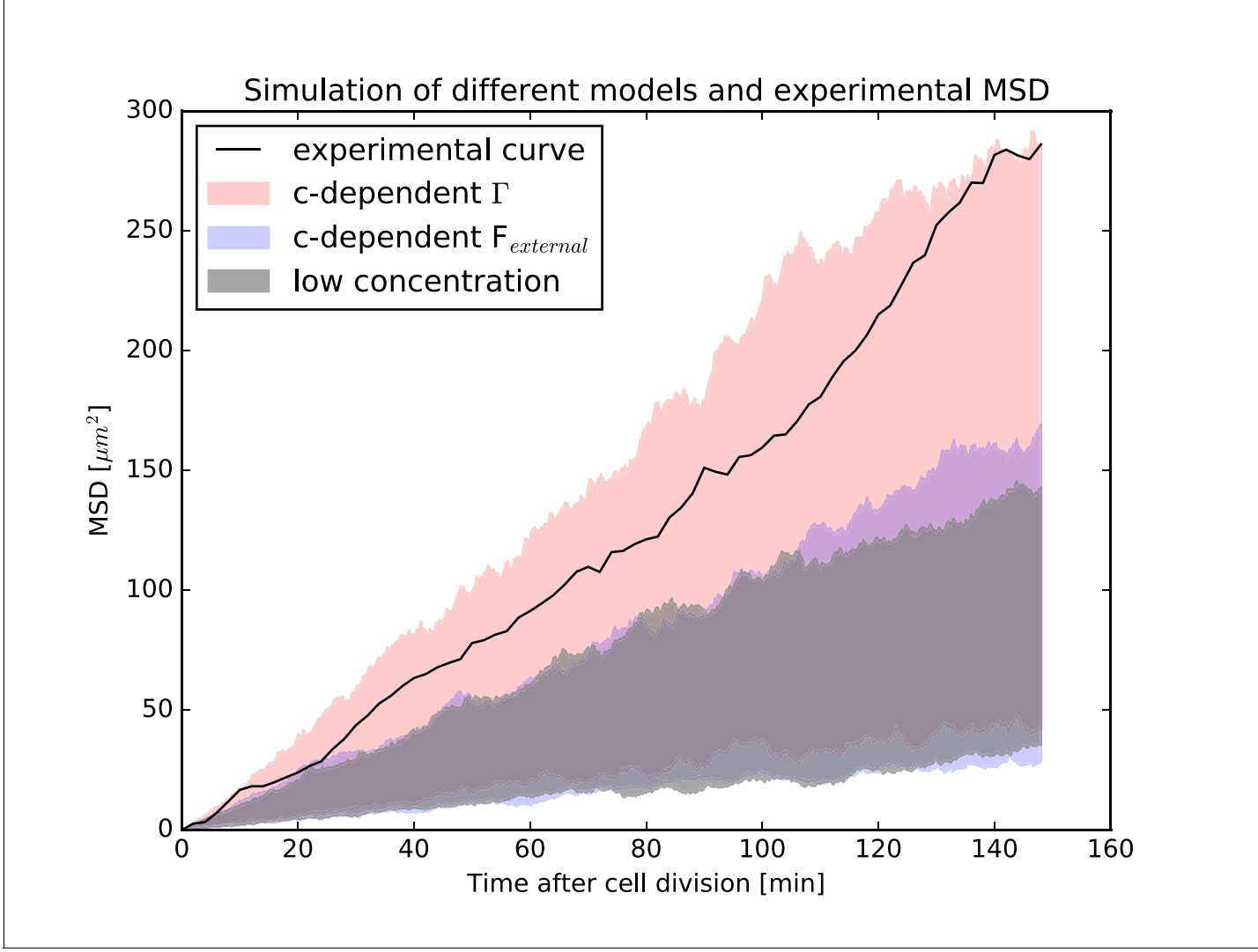

**Figure 8.** Mean-squared-displacement (MSD) of the first 40 nuclei that could be tracked beginning with cell division in the experiment. The black curve is the experimental MSD curve as a function of (cell-internal) time after cell division. The shaded areas represent the simulations of different models. In red is the model that assumes the effect of surrounding nuclei is due to a concentration dependence of the stochastic force (i.e. has a concentration-dependent $\Gamma$). In blue is the model that includes the effect of surrounding nuclei via an additional force $F_{\text{external}}$. In gray is the model for low nuclear concentration for comparison. In each case, the same 40 nuclei as the experiment have been simulated, taking their respective environment (i.e. the surrounding nuclear concentration) into account. In each simulation, the MSD curve was calculated as in the experiment. For each model, simulations were repeated 2500 times and the shaded areas represent the range of values covered by the individual resulting MSD curves for each model. The experimental MSD curve only agrees with the model assuming a concentration-dependent stochastic force.

a single-cell phenomenon. Instead, we can only interpret quantities such as MSD curves of nuclei undergoing IKNM correctly if we explicitly take the surrounding nuclei into account, even if there seems to be no direct energy transfer between nuclei, as shown from our experimental work (*Figure 3*). Second, the simulation results shown in *Figure 8* provide a means to distinguish between different ways in which the neighboring nuclei may act on a moving nucleus. As the experimental MSD curve only agrees with the model that assumes a concentration-dependent stochastic force, among those considered, the results indicate that cells are, in some manner, sensitive to the local nuclear concentration. As we have previously shown, the strength of this stochastic force is compatible with cytoskeletal transport. At high nuclear concentrations (i.e. when nuclei are packed close to the maximum possible packing density), as is the case closest to the apical surface of the retinal tissue, cells may recruit more molecular motors to transport nuclei away from this surface faster, leading to the concentration dependence of the stochastic force.

## Incubation temperature has direct effects on IKNM

The diffusion model may also address mechanistic questions about IKNM in retinas growing under varying experimental conditions. Zebrafish embryos are often grown at different temperatures to manipulate their growth rate (*Kimmel et al., 1995*; *Reider and Connaughton, 2014*), but it has been unclear how the nuclei in the retina behave at these different temperatures. To examine this issue, we grew the embryos at the normal temperature of 28.5 °C overnight and then incubated them at lower temperature (LT) of 25 °C or higher temperature (HT) of 32 °C during imaging. We could directly measure the change in average cell cycle length from experimental data and found that in HT, it is 205.5 min, while in LT, it is a much larger 532.78 min. We were then able to use these values in the model to investigate whether the change in temperature influences the processes that determine the effective diffusion constant of the nuclei. The resulting values for $D^*_{\mathrm{nonlin}}$ are summarized in *Table 1*. Based on these values, two-sided *t*-tests (see Materials and methods) confirmed that there is no significant difference between the *D*-values obtained from the two normal condition data sets. In contrast, *D*-values for the LT and HT data sets were significantly different from the normal ones, with $p \leq 0.01$. These results indicate, that aside from its effect on cell cycle length, incubation temperature is likely to influence IKNM directly by altering the mobility of nuclei, here represented by the effective diffusion constant *D*.

## Discussion

In this work, we have shown that high-density nuclear trajectories can be used to tease apart the possible physical processes behind the apparently stochastic movement of nuclei during interkinetic nuclear migration. First, we acquired these trajectories using long-term imaging and tracking of nuclei with high spatial and temporal resolution within a three-dimensional segment of the zebrafish retina. Analysis of speed and positional distributions of more than a hundred nuclei revealed a large degree of variability in their movements during G1 and S phases. Although this variability had been observed before, previous experiments had only considered sparsely labeled nuclei within an otherwise unlabeled environment (*Baye and Link, 2007*; *Norden et al., 2009*; *Leung et al., 2011*). Thus, our results provide an important account of the variability of IKNM on a whole tissue level. In effect, the variability in IKNM means that nuclear trajectories appear stochastic during the majority of the cell cycle. Previously, it had been suggested that the origins of this apparent stochasticity lay in the transfer of kinetic energy between nuclei in G2 exhibiting rapid apical migration to nuclei in G1 and S phases of the cell cycle, much as a person with an empty beer glass may nudge away other customers to get to the bar (*Norden et al., 2009*). However, we found no evidence for direct transfer of kinetic energy between nuclei and their immediate neighbors. Recently, *Shinoda et al., 2018* have also provided evidence that suggests direct collisions do not contribute to basal IKNM.

Another possibility is that the stochastic trajectories of G1 and S nuclei could be a result of nuclear crowding at the apical surface (*Miyata et al., 2014*), which, in effect, gives rise to a nuclear concentration gradient from the apical to the basal side of the tissue. This gradient is formed and sustained by nuclear divisions taking place exclusively at the apical surface. While the newly divided daughter nuclei are approximately 0.8 the size of M phase nuclei within the first 10 min after division, they increase in size in the following 10 min to become statistically indistinguishable from M phase nuclei. Thus, the difference in the nuclear density apicobasally is unlikely to be a direct result of variability in nuclear sizes during cell cycle. We confirmed the presence of a nuclear concentration gradient by calculating the nuclear concentration along the apicobasal dimension within the retinal tissue at various time points. Furthermore, to probe the source of the gradient, we treated the zebrafish retina with HU-AC to stop the cell cycle in S phase. While we observed the build-up of the nuclear concentration gradient over time in the control retina, the nuclear distribution flattened when cell division was inhibited with HU-AC treatment. Recent work indicates that only a small fraction of the apical tissue surface is occupied by mitotic cells at any given time (*Matejčić et al., 2018*). Nonetheless, even this small fraction consistently adds to the number of cells at the apical surface (*Figure 5A*) contributing to the observed evolving gradient shown in *Figure 6*.

These phenomenological similarities between IKNM and diffusion suggested a model that includes two key features: firstly, it focuses on the crowding of nuclei at the apical surface of the tissue, here included as the apical boundary condition. Secondly, in the nonlinear extension of the model, it incorporates a maximum possible nuclear concentration. This addition provided a striking

overall improvement to the fits to experimental data over periods of many hours. The resulting difference in the obtained $D$-values between the linear and nonlinear versions of our model can be understood heuristically when closely examining the difference between *Equations 1 and 9*. The latter introduces the new term $c_{\max}/(c_{\max} - c)$ which one could think of loosely as corresponding to an effective, concentration dependent diffusion constant $\tilde{D} = D c_{\max}/(c_{\max} - c)$. In general, $\tilde{D}$ will vary across the tissue thickness and, since $c$ is nonzero for most of the retinal tissue, $\tilde{D} > D$. Therefore, averaging across the retinal tissue, $\tilde{D}$ may actually be in very good agreement with the $D$-value found in the linear model. However, the linear model fails to describe the concentration-dependent nuclear mobility, which is successfully captured in the nonlinear model.

We made further use of the above correspondence between the linear and nonlinear models to obtain a microscopic interpretation of the particular value we obtained for $D^*_{\text{nonlin}}$, since both models converge into one another at $c \to 0$. The value of $D^*$ can neither be understood by assuming simple thermal diffusion of the nuclei, nor by simply including effects of membrane-hindered diffusion. Instead, it appears that both hindering and nonequilibrium driving forces have to be included, where nuclear mobility can be slowed-down due to the presence of the membrane and cytosolic composition and sped-up through active transport. Assuming membrane effects and active transport in a Langevin-type model for nuclei at low densities provided an estimate for the strength of the required transport forces, which is consistent with cytoskeletal transport of the nuclei throughout the cell cycle.

We then extended the Langevin-type model for individual nuclei to include the effects of high nuclear packing densities. The resulting models provided a possibility of exploring the properties of individual nuclear trajectories under conditions similar to those found in the experiments. Simulations using different models suggested that the effects of the dense nuclear packing influence the nuclear mobility by locally increasing the strength of the stochastic force. Importantly, the MSD curves obtained in the presence of crowding are essentially linear, even though the underlying dynamics are definitely nonlinear. This illustrates clearly the fact that the linearity of an MSD is not, by itself, particularly probative of the underlying diffusive dynamics.

The underlying processes causing IKNM during the G1 and S phases of the cell cycle in pseudostratified epithelia have been largely elusive. Several partially competing ideas have been put forward, ranging from the active involvement of cytoskeletal transport processes to passive mechanisms of direct energy transfer or movements driven by apical nuclear crowding (*Schenk et al., 2009*; *Tsai et al., 2010*; *Norden et al., 2009*; *Kosodo et al., 2011*). The fact that inanimate microbeads migrate much like nuclei during IKNM in the mouse cerebral cortex (*Kosodo et al., 2011*) suggests that active, unidirectional intracellular transport mechanisms are not directly responsible for these stochastic movements. Instead, we show that a passive diffusive process which takes steric interactions between nuclei into account produces an excellent representation of the time evolution of the actual nuclear distribution within the retinal tissue during early development. Consequently, our work builds on earlier models of apical crowding based on in silico simulations of IKNM (*Kosodo et al., 2011*). However, in contrast to earlier studies, we explicitly account for the dense nuclear packing within the zebrafish retina. Furthermore, we provide an interpretation for the general scale of the diffusion constant ($D \sim 0.1\ \mu\text{m}^2/\text{min}$) from microscopic considerations, similar to those used to relate random walks to diffusion (*Goldstein, 2018*). The results of these microscopic considerations strongly suggest that nuclei are moved by means of cytoskeletal transport throughout the entirety of the cell cycle. However, this transport appears not to be unidirectional but highly stochastic during basal IKNM.

Finally, an extension of the single nuclei equations to high concentrations and the results of stochastic simulations of nuclear trajectories suggest that the stochastic forcing of nuclei itself is concentration-dependent. On a microscopic scale, this can be interpreted, for example, under the assumption that cells can sense the nuclear packing density. If they recruited more molecular motors to areas where nuclei are particularly densely packed, the strength of the stochastic transport forces would be concentration-dependent. Nuclei would thus be transported away from areas of high nuclear packing faster. In addition to these microscopic considerations, our work reveals the importance of simple physical constraints imposed by the overall tissue architecture, which could not be explored in previous studies which tracked sparse nuclei, and thus lacked the means to explore the effect of such three-dimensional arrangements. Hence, we paid special attention to the spherical

shape of the retina and the concentration of nuclei in that space. Examining the evolution in distribution of nuclei over time unveils the importance of spatial restriction due to the curvature of the tissue. Additionally, the size of the nuclei in comparison to the tissue leads to the emergence of a maximum nuclear concentration which must be taken into account to model IKNM accurately.

By inhibiting cell cycle progression or changing temperature, we used the model to shed light on properties and mechanisms of the stochastic movements of nuclei during IKNM. From our results and previous studies, we know that cell cycle length is affected by change in incubation temperature (*Kimmel et al., 1995*; *Reider and Connaughton, 2014*). However, our results also indicate a significant influence of temperature on the mobility of nuclei and thus the underlying processes controlling their movement. This is reasonable in the light of our microscopic interpretations, which suggested that nuclei move due to cytoskeletal transport through the entire cell cycle in IKNM. The fact that the speed and dynamic properties of both the microtubule and actomyosin systems are temperature dependent may explain the changes in the diffusion constant that we see as a function of temperature (*Hartshorne et al., 1972*; *Hong et al., 2016*). In particular, as thermal diffusion is dependent on absolute temperature so the changes in temperature used in these experiments would have little effect on thermal diffusion. Furthermore, disparate observations seem to agree with such an interpretation. For example, a microtubule cage was observed around RPC nuclei (*Norden et al., 2009*) and myosin was also shown to surround these nuclei (*Leung et al., 2011*). Disruption of a microtubule motor (dynactin-1) functionality either by mutation (*Del Bene et al., 2008*) or introduction of a dominant negative allele (*Norden et al., 2009*) leads to a more basal positioning of nuclei and occasional bursts of basal movement. A conjecture consistent with these observations would be that during G1 and S phases actomyosin based forces push the nucleus basally, as also seen in the mouse telencephalon (*Schenk et al., 2009*), while microtubule motors push it apically. Finally, in G2 a concentration of myosin at the basal side of the nucleus leads to its rapid apical migration (*Leung et al., 2011*). However, a much closer examination of molecular mechanisms driving stochastic nuclear movements is required to better understand the connections between these phenomena, as we are far from understanding the nature of all the different forces involved in this process (*Kirkland et al., 2020*). Furthermore, the diffusion constant reported here reflects all types of nuclear movement during IKNM as it is derived from the changing nuclear concentration profile over time. It is not immediately clear how rapid apical migration contributes to this overall diffusion constant. Nonetheless, despite the large displacement during rapid apical migration at G2, this phase only accounts for about 8% of the cell cycle in RPCs (*Leung et al., 2011*). Therefore, the good agreement of our calculated diffusion constant with those previously reported in the literature for individual nuclei (*Leung et al., 2011*) suggests that the proposed model describes tissue-wide IKNM quite well. At the same time, it raises interesting new questions, such as how cells sense such concentrations and the mechanisms that increase the stochastic force on nuclear movement at higher concentrations.

The physiological consequences of nuclear arrangements and IKNM associated with all pseudostratified epithelia are not well understood. Our results provide a quantitative description of the stochastic distribution of the nuclei across the retina. This distribution has been implicated in stochastic cell fate decision making of progenitor cells during differentiation (*Clark et al., 2012*; *Baye and Link, 2007*; *Hiscock et al., 2018*). Our observations would fit with previous suggestions that a signalling gradient, such as Notch, exists across the retina and location-dependent exposure to it is important for downstream decision-making (*Murciano et al., 2002*; *Del Bene et al., 2008*; *Hiscock et al., 2018*; *Aggarwal et al., 2016*). Thus, our results not only have important implications for understanding the organization of developing vertebrate tissues, but may also provide a starting point for further exploration of the connection between variability in nuclear positions and cell fate decision making in neuroepithelia.

## Materials and methods

### Animals and transgenic lines

All animal works were approved by Local Ethical Review Committee of the University of Cambridge and performed in accordance with a Home Office project license PL80/2198. All zebrafish were maintained and bred at 26.5 ℃. All embryos were incubated at 28.5 ℃ before imaging sessions. At

10 hr post-fertilization (hpf), 0.003% phenylthiourea (PTU) (sigma) was added to the medium to stop pigmentation in the eye.

## Lightsheet microscopy

Images of retinal development for the main data set were obtained using lightsheet microscopy. Double transgenic embryos, Tg(bactin2:H2B-GFP::ptf1a:DsRed) were dechorionated at 24 hpf and screened positive for the fluorescent transgenic markers prior to the imaging experiment. The embryo selected for imaging was then embedded in 0.4% low gelling temperature agarose (Type VII, Sigma-Aldrich) prepared in the imaging buffer (0.3x Daniau's solution with 0.2% tricaine and 0.003% PTU [*Godinho, 2011*]) within an FEP tube with 25 μm thick walls (Zeus), with an eye facing the camera and the illumination light shedding from the ventral side. The tube was held in place by a custom-designed glass capillary (3 mm outer diameter, 20 mm length; Hilgenberg GmbH). The capillary itself was mounted vertically in the imaging specimen chamber filled with the imaging buffer. To ensure normal development, a perfusion system was used to pump warm water into the specimen chamber, maintaining a constant temperature of 28.5 °C at the location of the specimen.

Time-lapse recording of retinal development was performed using a SiMView light-sheet microscope (*Tomer et al., 2012*) with one illumination and one detection arm. Lasers were focused by Nikon 10x/0.3 NA water immersion objectives. Images were acquired with Nikon 40x/0.8 NA water immersion objective and Hamamatsu Ocra Flash 4.0 sCMOS camera. GFP was excited with scanned light sheets using a 488 nm laser, and detected through a 525/50 nm band pass detection filter (Semrock). Image stacks were acquired with confocal slit detection (*Baumgart and Kubitscheck, 2012*) with exposure time of 10 ms per frame, and the sample was moved in 0.812 μm steps along the axial direction. For each time point, two $330 \times 330 \times 250$ μm$^3$ image stacks with a 40 μm horizontal offset were acquired to ensure the coverage of the entire retina. The images were acquired every 2 min from 30 hpf to 72 hpf. The position of the sample was manually adjusted during imaging to compensate for drift. The two image stacks in the same time point were fused together to keep the combined image with the best resolution. An algorithm based on phase correlation was subsequently used to estimate and correct for the sample drift over time. The processing pipeline was implemented with MATLAB (MathWorks).

## Two photon microscopy

Images for the repetition data set and all other conditions were obtained using a TriM Scope II 2-photon microscope (LaVision BioTec). A previously established Tg(H2B-GFP) line, generated by injecting a DNA construct of H2B-GFP driven from the actin promoter (*He et al., 2012*), was used for all these experiments. Embryos were dechorionated and screened for expression of GFP at 24 hpf. An embryo was then embedded in 0.9% UltraPure low melting point agarose (Invitrogen) prepared in E3 medium containing 0.003% PTU and 0.2% tricaine. The agarose and embryo were placed laterally within a 3D printed half cylinder of transparent ABS plastic, 0.8 mm in diameter, attached to the bottom of a petri dish, such that one eye faced the detection lens of the microscope. The petri dish was then filled with an incubation solution of E3 medium, PTU, and tricaine in the same concentrations as above. For the experiment involving cell cycle arrest, hydroxyurea and aphidicolin (Abcam) were added to the incubation solution right before imaging, to a final concentration of 20 mM and 150 μM, respectively. The imaging chamber was maintained at a temperature of 25 °C, 28.5 °C, or 32 °C, as required, using a precision air heater (The Cube, Life Imaging Services).

Green fluorescence was excited using an Insight DeepSee laser (Spectra-Physics) at 927 nm. The emission of the fluorophore was detected through an Olympus 25x/1.05 NA water immersion objective, and all the signals within the visible spectrum were recorded by a sensitive GaAsP detector. Image stacks with step size of 1 μm were acquired with exposure time of 1.35 ms per line averaged over two scans. The images were recorded every 2 min for 10–15 hrs starting at 26–28 hpf. The same post-processing procedure for data compression and drift correction was used on these raw images as on those from lightsheet imaging.

## Obtaining experimental input values for the model

The radial coordinates $r_n$ of nuclei were calculated by subtracting $l_n$ from $a$, wherein $l_n$ is the distance from the center of a nucleus $n$ to the apical surface and $a$ is the distance from the center of the lens

to the apical surface. We estimated a total uncertainty of $\Delta r = \pm 3$ µm for each single distance measurement of $r_n$. This value is a result of uncertainty in detecting the center of the nucleus and in establishing the position of the apical surface.

Because each nuclear position has an error bar $\Delta r$, binning the data leads to an uncertainty in the bin count. In order to calculate this uncertainty, we considered the probability distribution of a nucleus' position. In the simplest case, this probability is uniform within the width of the positional error bar and zero elsewhere. The probability, $p_{n,\text{bin}}$, of finding a given nucleus $n$ within a given bin, is proportional to the size of the overlap of probability distribution and bin. It follows that the expectation value for the number of nuclei within a bin is given as $E(N_{\text{bin}}) = \sum_n p_{n,\text{bin}}$. Correspondingly, $\text{Var}(N_{\text{bin}}) = \sum_n p_{n,\text{bin}}(1 - p_{n,\text{bin}})$ is the variance of the number of nuclei within this bin. Thus, the error bar of the bin count is $\sigma_{y,\text{bin}} = \sqrt{\text{Var}(N_{\text{bin}})}$. The nuclear distribution profile $N(r,t)$ is not expected to be uniform or linear, therefore the expectation value $E(N_{\text{bin}})$ does not correspond to the number of nuclei at the center of the bin. Since the position of the expectation value is unknown a priori, it is still plotted at the center of the bin with an error bar denoting its positional uncertainty. Here we assume this error bar to be the square-root of the bin size $\Delta r_{\text{bin}}$, that is, $\sigma_{x,\text{bin}} = \sqrt{\Delta r_{\text{bin}}}$.

In order to obtain the experimental nuclear concentration profile $c(r,t)$, and its error bars, from the distribution of nuclei $N(r,t)$, the volume of the retina also has to be taken into account, since $c = N/V$. The total retinal volume within which nuclei tracking took place was estimated directly from the microscopy images. To this end, we outlined the area of observation in each image slice using the Fiji software and multiplied this area with the distance between successive images. Given the total volume, $V_{\text{total}}$, we proceeded to calculate the volume per bin, which depends on the radii at the inner and outer bin surfaces. In general, the volume of a spherical sector is $V_{\text{sector}} = \frac{1}{3}\Omega r_{\text{sector}}^3$, where $\Omega$ denotes the solid angle. Knowing the apical and basal tissue radii, $r = a$ and $r = b$, one can thus calculate $\Omega$ as $\Omega = 3V_{\text{total}}/(a^3 - b^3)$. This gives the volume of each bin as $V_{\text{bin}} = \frac{1}{3}\Omega\left(r_{\text{bin,outer}}^3 - r_{\text{bin,inner}}^3\right)$, where $r_{\text{bin,outer}}$ and $r_{\text{bin,inner}}$ denote the outer and inner radii of a bin, respectively. Similarly, we calculated the effective surface area $S$ through which the influx of nuclei occurs (see **Equation 3**) from the solid angle $\Omega$. This surface area is simply given as $S = \Omega a^2$.

To retrieve the average cell cycle time $T_P$ for each of the data sets, we used two different approaches. In the case of the main data set, sufficient number of nuclear tracks consisting of a whole cell cycle were present. Thus, we directly calculated the average cell cycle duration from these tracks. For the other data sets, we make use of the fact that the number of nuclei follows an exponential growth law depending on $T_P$ (see **Equation 2**). Knowing the initial number of tracked nuclei $N_0$ for each data set, we obtained $T_P$ from fitting the following equation to the number of nuclei as a function of time in a log-lin plot: $\ln N(t) = \ln N_0 + t/\tau = \ln N_0 + (\ln 2/T_P)t$. Then $T_P$ was deduced from the slope of this fit.

In order to determine the maximum nuclear concentration $c_{\text{max}}$ for the nonlinear model, we first randomly selected 100 nuclei from our dataset of tracked nuclei and measured the size of their longest diameter in both XY and YZ planes. From these measurements, we established that the size of the principal semi-axis of each nucleus is likely to lie in the range of about 3 µm to 5 µm, where the nuclear shape is regarded to be ellipsoidal. This led to the range of possible maximum concentrations $c_{\text{max}}$, although we did not measure the precise nuclear volume. The lower limit for the nuclear volume is set by the volume of a sphere of radius 3 µm, the upper limit by a sphere of radius 5 µm. Taking into account the maximum possible packing density of nuclei, which for aligned ellipsoids is the same as that of spheres (**Donev et al., 2004**), $\pi/(3\sqrt{2}) \approx 0.74$, we obtained a range of $1.41 \times 10^{-3}$ $\mu\text{m}^{-3} \leq c_{\text{max}} \leq 6.55 \times 10^{-3}$ $\mu\text{m}^{-3}$.

## Obtaining the initial condition

We determined the prefactors $h_i$ from the experimental nuclear distribution at the start of the experiment, $c_{\text{exp}}(\xi, 0)$. For convenience, we chose to determine first $\widetilde{h}_i = h_i + \alpha_i f_0/(\sigma + \lambda_i^2)$ and then obtained $h_i$ by subtracting $\alpha_i f_0/(\sigma + \lambda_i^2)$ from the results. The $\widetilde{h}_i$ can be calculated from the data, using **Equation 6** for $s = 0$, as

$$\widetilde{h}_i = \sum_m \xi_m^2 H_i(\xi_m) c_{\text{exp}}(\xi_m, 0)\Delta\xi_m - \frac{f_0}{1-\rho}\int_\rho^1 \xi^2 H_i(\xi)\left(\frac{1}{2}\xi^2 - \rho\xi + g_0\right)d\xi, \tag{17}$$

where $m$ denotes the $m$-th binned data point, $\xi_m$ its position and $\Delta\xi_\mathrm{m}$ the width of bin $m$. As in *Equation 6*, the index $i$ denotes the $i$-th eigenfunction or -mode.

## The concentration profile in the nonlinear model

The non-linear concentration profile was determined numerically from the same initial condition as used for the linear model, *Equation 6*, at $s = 0$ with $\widetilde{h}_i$ as in *Equation 17*. Time evolution of the initial condition, according to *Equation 9*, was performed using the pdepe solver in MATLAB.

## Fitting the model

The range of sizes of the nuclear principal semi-axes was used to determine the range of data to be included in our fits. Any data closer than 3 μm to 5 μm from the apical or basal tissue surfaces was not taken into account for fitting because the center of a nucleus cannot be any closer to a surface than the nuclear radius. Thus, all data collection very close to the apical or basal tissue surfaces must have been due to the above-mentioned measurement uncertainties $\Delta r$.

In principle, the full solution for $c(\xi, s)$ is composed of infinitely many modes. However, in practice, we truncated this series and only included the first eight modes in our fits. This is due to the fact that we have a finite set of data points, so adding too many modes could lead to over-fitting. Fits with a wide range of numbers of modes were found to result in the same optimal $D$-values.

For fitting, we first rescaled the data in accordance with the non-dimensionalization of the theoretical variables $r$ and $t$ (see *Equation 5*). Thus we obtain $c_\mathrm{exp}(\xi, s)$ from $c_\mathrm{exp}(r, t)$. Then both models were fitted to the experimental data using a minimal-$\chi^2$ approach. The goodness of fit parameter $\chi^2 = \sum_m \left(c_\mathrm{exp}(\xi, s) - c(\xi, s)\right)^2 / \sigma_m^2$, where $\sum_m$ denotes the summation over all bins $m$. Since binning resulted in uncertainties $\sigma_{y,\mathrm{bin}}$ and $\sigma_{x,\mathrm{bin}}$ in the $y$- and $x$-directions, both had to be taken into account when calculating $\sigma_m$ and $\chi^2$. The combined contribution of $x$- and $y$- uncertainties is: $\sigma_m^2 = \sigma_{y,m}^2 + \sigma_{y,\mathrm{indirect},m}^2$ with $\sigma_{y,\mathrm{indirect},m} = \sigma_{x,m}(\mathrm{d}c(\xi, s)/\mathrm{d}\xi)|_{\xi = \xi_m}$ (*Bevington and Robinson, 2003*). In our fits, the value $\chi^2$ was calculated for a large range of possible diffusion constants $D$, from $D = 0.01$ μm²/min to $D = 10$ μm²/min. By finding the value of $D$ for which $\chi^2$ became minimal for a given data set and time point, we established our optimal fit.

The minimal-$\chi^2$ approach furthermore enabled us to determine the optimal binning width $\Delta r_\mathrm{bin}$ or $\Delta\xi_\mathrm{bin}$ and width of data exclusion for the fits. In order to do so, fits of the normal data set were performed for different data binning widths and exclusion sizes of 3 μm to 5 μm. For each of these fits the $\chi^2$-value and the number of degrees of freedom $\nu$, that is, the number of data points minus the number of free fit parameters (here number of data points minus 1), were registered. From $\chi^2$ and $\nu$, we calculated the reduced $\chi^2$ value, $\chi_\nu^2 = \chi^2/\nu$ (*Bevington and Robinson, 2003*). Using $\nu$ and $\chi_\nu^2$, the probability $P_\chi(\chi^2; \nu)$ of exceeding $\chi^2$ for a given fit can be estimated, which should be approximately 0.5 (*Bevington and Robinson, 2003*). Therefore, we found our optimal data binning width of 3 μm to 4 μm as the width that resulted in a $P_\chi(\chi^2; \nu)$ as close to 0.5 as possible for all the different time points when fitting the nonlinear model. The exact choice of exclusion width was found not to influence the fitting result for the nonlinear model.

In addition to finding the optimal $D$-value for individual time points, we also modified the minimal-$\chi^2$ routine to find the value of $D$ that fits a whole data set (i.e. all time points simultaneously) in the best possible way. In order to do so, we summed the $\chi^2$-values obtained for each $D$ over all time points, in this way producing a $\sum_t \chi^2(D)$-curve. The minimum of this curve indicates $D^*$ for the whole time series. Furthermore, dividing $\sum_t \chi^2(D)$ by the number of time points included in the optimization yields an average $\chi^2$- and reduced $\chi^2$-value corresponding to this $D^*$. In addition, the width of this time averaged curve at $\chi^2 = \chi_\mathrm{min}^2 + 1$ indicates the standard deviation of the optimal $D$-value, $\sigma_D$. By approximating the minimum with a quadratic curve, we obtain an estimate for this standard deviation as $\sigma_D = \Delta_D\sqrt{2\left(\chi_{D^*-\Delta_D}^2 - 2\chi_{D^*}^2 + \chi_{D^*+\Delta_D}^2\right)}$ (*Bevington and Robinson, 2003*) where $\Delta_D$ is the step size between individual fitted $D$-values, here $\Delta_D = 0.01$ μm²/min. Lastly, based on the average reduced $\chi^2$-values, we also compared several $c_\mathrm{max}$-values for each data set to find the fit with probability $P_\chi(\chi^2; \nu)$ the closest to 0.5 in each case.

All fits were performed using custom MATLAB routines. Horizontal error bars were plotted using the function herrorbar (*van der Geest, 2006*).

## Nuclear radius for interpretation of *D*

The average nuclear radius used to calculate the friction coefficient and thermal diffusion coefficient of IKNM nuclei was the radius corresponding to the maximum concentration $c_{\mathrm{max}}$ obtained from the fitting procedure.

## Experimental nuclear birth times and mean-squared-displacement curve

Among all the nuclei tracked in the experiments, we selected those nuclei where tracking data was available beginning right from cell division and also over a sufficiently long period of time to cover a substantial part of the cell cycle (at least 75 time steps, i.e. 150 min). For these nuclei, we extracted their respective birth times within the experiment from the full tracks and sorted the nuclei accordingly. The first 40 nuclei were chosen for further analysis, as these were nuclei with a minimum of 150 min of tracking data completely within the first 200 min of experiments, corresponding to the time frame used for *D*-optimization in the non-linear diffusion model. The exact distribution of their birth times was stored for use in the individual nuclei simulations.

Further, the nuclear tracks of the chosen 40 nuclei were transformed from being a function of experimental time to being a function of cell cycle time by simply subtracting a nucleus individual birth time from the experimental time for each step of its tracking data. Then the experimental mean squared displacement curve was calculated from the so obtained cell-cycle-dependent tracks.

## Calculation of the shapes of retinal cell shapes

Here, we give more information on the numerical calculation of cell shapes. Further details can be found elsewhere (*Herrmann, 2020*). Minimization of the elastic energy (*Equation 12*) leads to the equilibrium condition on the shape, expressed in terms of the mean curvature $\mathcal{H}$ and the Gaussian curvature $\mathcal{K}$ (*Zhong-can and Helfrich, 1989*),

$$-\gamma \mathcal{H} + 2\kappa \left( \mathcal{H}^3 - \mathcal{K}\mathcal{H} \right) + \kappa \Delta \mathcal{H} = 0, \tag{18}$$

where, for an axisymmetric shape $\delta(z)$,

$$\mathcal{K} = -\frac{\delta_{zz}}{\delta \left( 1 + \delta_z^2 \right)^2} \tag{19}$$

and $\Delta$ is the Laplacian operator,

$$\nabla^2 = \frac{1}{\delta \sqrt{1 + \delta_z^2}} \frac{\partial}{\partial z} \left( \frac{\delta}{\sqrt{1 + \delta_z^2}} \frac{\partial}{\partial z} \right). \tag{20}$$

The resulting shape equation is fourth order in *z*-derivatives and thus requires four boundary conditions. Given the symmetry of the system, we solve for the shape in the left half of the domain $z = (0, L/2)$ and impose $\delta(0) = \delta_a$ and $\delta_z(0) = 0$ at the apical surface. Imposing boundary conditions like $\delta(L/2) = \mathfrak{R}$ and $\delta_z(L/2) = 0$ at the top of the nucleus usually leads to solutions that are incompatible with the presence of the nucleus (the resulting membrane shapes would cut through the nucleus). Therefore, we further divide the domain $z = (0, L/2)$ into a region away from the nucleus and a region where the membrane is in close contact with it. In the latter region, we assume the membrane to be bent into a spherical arc around the nucleus, leaving a small equilibrium gap as estimated by *Daniels, 2019*. The contact point $z_{\mathrm{contact}}$ between the two regions is adjusted until the membrane radius and its derivative are continuous through the contact point. The membrane shape away from the nucleus is then found using the MATLAB bvp5c solver. As can be seen from energy minimization using (*Equation 12*), the solution in each case turns out to be the one for which $z_{\mathrm{contact}}$ has been chosen such that the resulting $\mathcal{H}$ in $z \in [0, z_{\mathrm{contact}}]$ is equal to $\mathcal{H}_{\mathrm{circle}} = -1/\mathfrak{R}_{\mathrm{tube}}$ for $z \to z_{\mathrm{contact}}$, where $\mathfrak{R}_{\mathrm{tube}}$ is the radius of the membrane arc around the nucleus.

### Simulations of individual nuclear trajectories

Simulations of nuclear trajectories for each of the three Langevin-type models were performed using a custom Python 3 routine. Time discretization of the stochastic differential equations was achieved via the Euler-Maruyama method. Simulations were performed using 0.2 min time steps and were checked against those with smaller time steps to ensure that this choice was sufficiently small.

In each run of a simulation, 40 nuclei were simulated and their birth times within the simulation were chosen to be the same birth times as those obtained from the nuclei within the experiments. Each nucleus was simulated for a total of 150 min, corresponding to the chosen experimental data. The value for the diffusion constant in these simulations was set to be the previously obtained value $D^*_{\mathrm{nonlin}}$. For simulations with nuclear concentration-dependent Langevin equations, $c_{\mathrm{max}}$ and the average nuclear concentration field $c(r, t)$ were similarly extracted from the results of the previous fits using the non-linear diffusion equation. Herein, $c(r, t)$ was provided for each time step of the simulation. As $c(r, t)$ can only be provided for discrete spatial coordinates $r$ but the Langevin-type simulations were continuous in the spatial coordinate $r$, $c$ was averaged over the values at the two closest spatial points whenever a nucleus' position did not exactly coincide with a point where the value for $c$ was provided.

The resulting simulated nuclear trajectories were treated in the same way as the experimentally obtained ones. That is, the nuclei's birth times were subtracted from the trajectories to obtain cell cycle dependent tracks. Then, the mean squared displacement curve was calculated from the resulting set.

For each model, the same simulation was repeated 2500 times to obtain the range of distributions of the resulting mean squared displacement curves. For each cell cycle time step, the minimum and maximum of the mean squared displacement values out of all 2500 repetitions were calculated to obtain the areas depicted in *Figure 8*.

### t-tests

To compare results between data sets, the values $D^*$ and corresponding $\sigma_D$ from the overall fits were considered. It should be noted that these values were not obtained by averaging several data sets of the same experimental condition but instead each value results from one data set only. However, the sample size for each data set was set to 100 because 100 time points were taken into account for each overall optimization. These time points might not be completely uncorrelated, limiting the predictive power of the *t*-test. Two sided tests, specifically unequal variances *t*-test, also known as Welch's *t*-test, (*Precht and Kraft, 2015*), were performed in order to determine whether samples differ significantly from each other.

### Description of Matlab files

*Source code files 1*, *2*, *3*, *4*, *5*, *6* are Matlab files containing the tracking data, as follows.

## Acknowledgements

We thank Kevin O'Holleran and Martin Lenz at Cambridge Advanced Imaging Centre for their help and support in imaging zebrafish retinas. We also thank Oliver Y Feng, Timothy J Pedley, Michael E Cates and Salvatore Torquato for helpful advice and input. This work was supported by the Cambridge Wellcome Trust PhD Programme in Developmental Biology, the Cambridge Commonwealth, European and International Trust, and Natural Sciences and Engineering Research Council of Canada (AA); the Engineering and Physical Sciences Research Council, a Helen Stone Scholarship at the University of Cambridge through the Cambridge Commonwealth, European and International Trust, and the Cambridge Philosophical Society (AH); Established Career Fellowship EP/M017982/1 from the Engineering and Physical Sciences Research Council (REG); and Wellcome Trust Investigator Award (SIA 100329/Z/12/Z) (WAH).

# Additional information

## Competing interests

Raymond E Goldstein: Reviewing editor, *eLife*. The other authors declare that no competing interests exist.

## Funding

| Funder | Grant reference number | Author |
| --- | --- | --- |
| Engineering and Physical Sciences Research Council | EP/M017982/1 | Anne Herrmann Raymond E Goldstein |
| Wellcome Trust | 100329/Z/12/Z | William A Harris |
| Cambridge Commonwealth, European and International Trust | | Afnan Azizi Anne Herrmann |
| Natural Sciences and Engineering Research Council of Canada | | Afnan Azizi |
| Wellcome Trust | Cambridge Wellcome Trust PhD Programme in Developmental Biology | Afnan Azizi |
| Cambridge Philosophical Society | | Anne Herrmann |

The funders had no role in study design, data collection and interpretation, or the decision to submit the work for publication.

## Author contributions

Afnan Azizi, Conceptualization, Resources, Data curation, Software, Formal analysis, Validation, Investigation, Visualization, Methodology, Writing - original draft, Writing - review and editing; Anne Herrmann, Conceptualization, Resources, Data curation, Software, Formal analysis, Investigation, Visualization, Methodology, Writing - original draft, Writing - review and editing; Yinan Wan, Conceptualization, Resources, Data curation, Software, Formal analysis, Validation, Investigation, Methodology; Salvador JRP Buse, Software, Investigation; Philipp J Keller, Conceptualization, Supervision, Funding acquisition, Investigation, Methodology, Project administration; Raymond E Goldstein, Conceptualization, Formal analysis, Supervision, Validation, Investigation, Methodology, Writing - original draft, Project administration, Writing - review and editing; William A Harris, Conceptualization, Formal analysis, Supervision, Funding acquisition, Validation, Investigation, Methodology, Writing - original draft, Project administration, Writing - review and editing

## Author ORCIDs

Afnan Azizi (ID) https://orcid.org/0000-0002-3288-9612
Anne Herrmann (ID) https://orcid.org/0000-0003-1745-7499
Philipp J Keller (ID) http://orcid.org/0000-0003-2896-4920
Raymond E Goldstein (ID) https://orcid.org/0000-0003-2645-0598
William A Harris (ID) https://orcid.org/0000-0002-9995-8096

## Decision letter and Author response

Decision letter https://doi.org/10.7554/eLife.58635.sa1
Author response https://doi.org/10.7554/eLife.58635.sa2

## Additional files

### Supplementary files

• Source code 1. All track information for main data set. The fields within this data structure are: fullTr: n x six matrix where n is the number of all nuclei within one lineage. In this matrix: column 1 - object ID at each time point; columns 2–4 - (x,y,z); column 5 - parent ID (=ID from previous time point); column 6 - time point (2 min per time point). indTracks: Separated, "individual' tracks of a lineage where each track starts at t = 0 or a division and ends at t = end or another division. Each cell in the array is a track and contains a matrix with only nuclei of that track (same columns as fullTr). distVel: The distance and velocity vector for each nucleus at each time point. Each row corresponds to distance and velocity of the nucleus going from that time point to the next. Each cell corresponds to the same one in indTracks. In the matrices: column 1 - time point; column 2 - distance (magnitude of velocity vector); column 3 - cumulative distance; columns 4–6 - velocity vector (x,y,z) corrIndTracks and corrDistVel: The exact same as indTracks and distVel, but corrected for drift. Correction was carried out by calculating the sum of all displacement vectors at each time point and then changing all position vectors in indTracks to make that sum zero, which gave us corrIndTracks. That was then used to calculate corrDistVel. As for the second column of cells in corrIndTracks, it relates to the neighbours of each nucleus. Each element of a cell in the second column stores the address of neighbours of the corresponding nucleus in the first column (address matrix: column 1 - Tracks row index; column 2 - corrIndTracks row index; column 3 - object row index). apBasProj: The projection of velocity vectors onto the calculated normals to the apical surface. In this matrix: column 1–3 - apicobasal velocity vector (x,y,z); column 4 - magnitude of the vector; column 5 - direction (negative = apical and positive = basal). lateralProj: The component of the velocity vectors perpendicular to the apicobasal one. Same columns as in apBasProj without column 5.

• Source code 2. Set of distances of all nuclei from the apical surface for each time point for the normal condition data set (this was originally extracted from Tracks.mat but used in this way for all the analysis of nuclear distribution or concentration). The time points are every 2 min (including t = 0 min in the first column) except for the HUAC data set where they are every 40 min (including t = 0 min which is 120 min after drug treatment).

• Source code 3. Set of distances of all nuclei from the apical surface for each time point for the high temperature data set. The time points are every 2 min (including t = 0 min in the first column) except for the HUAC data set where they are every 40 min (including t = 0 min which is 120 min after drug treatment).

• Source code 4. Set of distances of all nuclei from the apical surface for each time point for the low temperature data set. The time points are every 2 min (including t = 0 min in the first column) except for the HUAC data set where they are every 40 min (including t = 0 min which is 120 min after drug treatment).

• Source code 5. Set of distances of all nuclei from the apical surface for each time point for the repeat normal condition data set. The time points are every 2 min (including t = 0 min in the first column) except for the HUAC data set where they are every 40 min (including t = 0 min which is 120 min after drug treatment).

• Source code 6. Set of distances of all nuclei from the apical surface for each time point for the HUAC treatment data set. The time points are every 2 min (including t = 0 min in the first column) except for the HUAC data set where they are every 40 min (including t = 0 min which is 120 min after drug treatment).

• Transparent reporting form

### Data availability

All data generated or analysed during this study are included in the manuscript and supporting files.

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

## Appendix 1

## Full solution of the linear diffusion equation

After rescaling space and time as in *Equation 5* and introducing $\rho = b/a < 1$, *Equation 1* and the boundary conditions *Equation 3* and *Equation 4* read

$$\frac{\partial c(\xi,s)}{\partial s} = \frac{1}{\xi^2}\frac{\partial}{\partial \xi}\left(\xi^2 \frac{\partial c(\xi,s)}{\partial \xi}\right),$$

$$\left.\frac{\partial c(\xi,s)}{\partial \xi}\right|_{\xi=1} = f_0 e^{\sigma s} = f(s) \qquad \text{and} \qquad \left.\frac{\partial c(\xi,s)}{\partial \xi}\right|_{\xi=\rho} = 0, \tag{21}$$

where we have defined $f_0 = aN_0/DS\tau$ and $\sigma = a^2/D\tau$. We transform this homogeneous differential equation with inhomogeneous boundary conditions into the problem of solving an inhomogeneous differential equation with homogeneous boundary conditions by writing $c(\xi,s)$ as a sum of two contributions,

$$c(\xi,s) = \phi(\xi,s) + \psi(\xi,s), \tag{22}$$

where we require $\phi(\xi,s)$ to satisfy the inhomogeneous boundary conditions

$$\left.\frac{\partial \phi(\xi,s)}{\partial \xi}\right|_{\xi=1} = f_0 e^{\sigma s} \qquad \text{and} \qquad \left.\frac{\partial \phi(\xi,s)}{\partial \xi}\right|_{\xi=\rho} = 0. \tag{23}$$

These conditions are satisfied if $\phi(\xi,s)$ has the form

$$\phi(\xi,s) = \frac{1}{1-\rho}\left(\frac{1}{2}\xi^2 - \rho\xi + g_0\right)f_0 e^{\sigma s}. \tag{24}$$

where $g_0$ is a constant of integration to be determined later. The remaining problem to solve for $\psi(\xi,s)$ is

$$\frac{\partial \psi(\xi,s)}{\partial s} = \frac{1}{\xi^2}\frac{\partial}{\partial \xi}\left(\xi^2 \frac{\partial \psi(\xi,s)}{\partial \xi}\right) + \frac{f_0 e^{\sigma s}}{1-\rho}\left(3 - \frac{2\rho}{\xi} - \sigma\left(\frac{1}{2}\xi^2 - \rho\xi + g_0\right)\right), \tag{25}$$

with homogeneous boundary conditions

$$\left.\frac{\partial \psi(\xi,s)}{\partial \xi}\right|_{\xi=1} = 0 \qquad \text{and} \qquad \left.\frac{\partial \psi(\xi,s)}{\partial \xi}\right|_{\xi=\rho} = 0. \tag{26}$$

We can further write $\psi(\xi,s)$ as the sum of two contributions,

$$\psi(\xi,s) = \psi_h(\xi,s) + \psi_p(\xi,s), \tag{27}$$

where $\psi_h$ is the general solution of the homogeneous problem

$$\frac{\partial \psi_h(\xi,s)}{\partial s} = \frac{1}{\xi^2}\frac{\partial}{\partial \xi}\left(\xi^2 \frac{\partial \psi_h(\xi,s)}{\partial \xi}\right),$$

$$\left.\frac{\partial \psi_h(\xi,s)}{\partial \xi}\right|_{\xi=1} = 0 \qquad \text{and} \qquad \left.\frac{\partial \psi_h(\xi,s)}{\partial \xi}\right|_{\xi=\rho} = 0, \tag{28}$$

and $\psi_p$ is a particular solution of the full inhomogeneous problem *Equation 26*. The full solution of the homogeneous problem is given as a series of linearly independent eigenfunctions, each of the form

$$e^{-\lambda^2 s}W(\xi) = e^{-\lambda^2 s}\left(A\frac{\sin\lambda\xi}{\xi} + B\frac{\cos\lambda\xi}{\xi}\right), \tag{29}$$

where the eigenvalues $\lambda$ can be found from simultaneous solution of the boundary conditions,

$$A(\lambda \cos \lambda - \sin \lambda) - B(\lambda \sin \lambda + \cos \lambda) = 0$$

$$A\left(\frac{\lambda \cos \lambda \rho}{\rho} - \frac{\sin \lambda \rho}{\rho^2}\right) - B\left(\frac{\lambda \sin \lambda \rho}{\rho} + \frac{\cos \lambda \rho}{\rho^2}\right) = 0, \tag{30}$$

which yields the transcendental relation

$$\tan \lambda (1 - \rho) = \frac{\lambda (1 - \rho)}{\lambda^2 \rho + 1}, \tag{31}$$

for which each eigenvalue $\lambda_i$ is a solution corresponding to one of the linearly independent eigenfunctions (only $\lambda_i > 0$ need to be taken into account). We can further deduce from the *Equation 30* that $B_i = \beta_i A_i$, where

$$\beta_i = \frac{\lambda_i \cos \lambda_i - \sin \lambda_i}{\lambda_i \sin \lambda_i + \cos \lambda_i}, \tag{32}$$

and we normalize the obtained expression for $W_i(\xi)$ from *Equation 29*

$$H_i(\xi) = \frac{1}{Y_i}\left(\frac{\sin \lambda_i \xi}{\xi} + \beta_i \frac{\cos \lambda_i \xi}{\xi}\right), \tag{33}$$

with

$$Y_i^2 = \frac{1}{2}(1 - \rho)(1 + \beta_i^2) - \frac{1}{4\lambda_i}(\sin 2\lambda_i - \sin 2\lambda_i \rho)(1 - \beta_i^2) + \frac{\beta_i}{\lambda_i}(\sin^2 \lambda_i - \sin^2 \lambda_i \rho). \tag{34}$$

Thus, the homogeneous solution $\psi_h$ is

$$\psi_h = \sum_{i=1}^{\infty} h_i H_i(\xi) e^{-\lambda_i^2 s}, \tag{35}$$

with prefactors $h_i$ to be determined from the initial condition.

In order to find a particular solution of the inhomogeneous problem, we first rewrite *Equation 26* as

$$\frac{\partial \psi(\xi, s)}{\partial s} - \frac{1}{\xi^2}\frac{\partial}{\partial \xi}\left(\xi^2 \frac{\partial \psi(\xi, s)}{\partial \xi}\right) = \mathcal{R}(\xi, s). \tag{36}$$

Now, we express $\mathcal{R}(\xi, s)$, as well as the unknown inhomogeneous solution $\psi_p(\xi, s)$ in terms of the normalized eigenfunctions $H(\xi, s)$ of the homogeneous problem,

$$\mathcal{R}(\xi, s) = \sum_{i=1}^{\infty} R_i(s) H_i(\xi), \tag{37}$$

and

$$\psi_p(\xi, s) = \sum_{i=1}^{\infty} C_i(s) H_i(\xi). \tag{38}$$

Substituting these forms into *Equation 36*, and noting that each term in the series must vanish separately we obtain

$$\frac{\partial C_i(s)}{\partial s} + \lambda_i^2 C_i(s) - R_i(s) = 0. \tag{39}$$

From the form of $\mathcal{R}(\xi, s)$ it follows that $R_i(s) = \alpha_i f_0 e^{\sigma s}$ with some purely numerical prefactors $\alpha_i$, so we expect $C_i(s) \propto p_i e^{\sigma s}$ and find

$$p_i = \frac{\alpha_i f_0}{\sigma + \lambda_i^2}. \tag{40}$$

Finally, we determine the $\alpha_i$ by reconsidering *Equation 37*. We multiply both sides by $\xi^2 H_j(\xi)$,

where $H_j(\xi)$ is one specific but arbitrary eigenfunction of the homogeneous problem, and then integrate over the whole volume $V$. By the orthogonormality of these eigenfunctions, we obtain

$$\alpha_j = \int \frac{1}{1-\rho}\left(3 - \frac{2\rho}{\xi} - \sigma\left(\frac{1}{2}\xi^2 - \rho\xi + g_0\right)\right)\xi^2 H_j(\xi)d\xi, \tag{41}$$

and all the $\alpha_i$ can be calculated explicitly. Thus, the full solution of the linear problem is

$$c(\xi,s) = \sum_{i=1}^{\infty}\left(h_i e^{-\lambda_i^2 s} + \frac{\alpha_i f_0}{\sigma + \lambda_i^2}e^{\sigma s}\right)H_i(\xi) + \frac{1}{1-\rho}\left(\frac{1}{2}\xi^2 - \rho\xi + g_0\right)f_0 e^{\sigma s}. \tag{42}$$

The constant $g_0$ can now be calculated from the requirement that $\int c(\xi, s = 0)dV = N_0$. Here, we make use of the fact that $\int H_i(\xi)\xi^2 d\xi = 0$ if $\lambda_i$ satisfies *Equation 31*, thus

$$g_0 = \frac{(1-\rho)/\sigma - \frac{1}{10} + \frac{1}{4}\rho + \frac{1}{10}\rho^5 - \frac{1}{4}\rho^5}{\frac{1}{3}(1-\rho^3)}. \tag{43}$$

