## [Decision Letter]

**Acceptance summary:**

This paper the authors describe the process of interkinetic nuclear migration tracking for the first time all nuclei within a defined volume of neuroepithelium. Combining this experimental approach with theoretical analysis they showed that this important and well-studied phenomenon in developmental biology can be described quantitatively using the physical principles that underlie diffusion.

**Decision letter after peer review:**

[Editors’ note: the authors submitted for reconsideration following the decision after peer review. What follows is the decision letter after the first round of review.]

Thank you for submitting your work entitled "Interkinetic nuclear migration in the zebrafish retina as a diffusive process" for consideration by *eLife*. Your article has been reviewed by a Senior Editor, a Reviewing Editor, and three reviewers. The reviewers have opted to remain anonymous.

Our decision has been reached after consultation between the reviewers. Based on these discussions and the individual reviews below, we regret to inform you that your work will not be considered further for publication in *eLife*.

As you see from the comments below the reviewers highlight the main strength of your paper being the presentation of a comprehensive physical description of the process of IKNM.

They also feel that from a conceptual point the advances are mostly incremental since that basal migration is a diffusive process was substantially shown before by Norden et al. (2009) and Leung et al. (2011). Unfortunately the general consensus was that the present study does not provides anything conceptually new as it is mainly descriptive.

Reviewer #1:

The manuscript by Harris, Goldstein and colleagues use a combination of experiments and theory to describe retinal IKNM as a diffusive process. The movement of nuclei during G1 and S phase have previously been suggested to be passive and to be caused by apical crowding. The description of this process as a diffusive process is thus a logical extension of these previous proposals. That said, the strength of the present study is certainly more in providing a comprehensive physical description of the process rather than coming up with an entirely new concept.

The only major criticism I have is that the study is mainly descriptive with very limited functional interventions. Blocking S phase with HUA and changing temperature can have multiple side-effects that make the interpretations of the resulting phenotype more difficult. There have been more elegant genetic experiments in the past for interfering with apical crowding – for instance by knocking down TAG1 or overexpressing Wnt3 – that could have been used to challenge the predictions from the diffusion theory.

Collectively, I think the study in its present form would in principle be suitable for publication although some more experimental work to address theoretical predictions would be preferable.

Reviewer #2:

In this manuscript the authors combine timelapse microscopy, cell tracking, and theory to develop a model of interkinetic nuclear migration in the pseudostratified epithelium of the developing zebrafish retina. It is argued that a theory based on diffusion in a concentration gradient can model the observed data.

This paper fits in with several previous works which have used the zebrafish retina as a model for quantitatively looking at IKNM including ones that have measured the effective cellular diffusion constant (Leung et al., 2011) and argued that the movement is diffusive.

My major concerns are with the assumptions of the diffusion model:

1) What causes the gradient? In the model it is argued that there is a diffusion gradient because cells are added at the apical surface because that is where mitosis takes place. However, when a cell divides, the daughters are typically half the volume of their mother and then double in size over the course of interphase. Daughter nuclei after division have half the DNA of their mother just before division so they are presumably smaller also, and would then grow during S-phase. It is important to measure nuclear volume as a function of cell cycle phase before asserting that apical mitosis is creating the gradient. It seems apically directed movement of cells in late G2 could also create a concentration gradient although this is a small net movement that could be quickly relaxed.

2) What causes the movement? The Abstract claims to "uncover the physical process" of IKNM which I don't think it does. It uses the same mathematical framework as would be used to study molecular diffusion as a physical process, but I would argue this is a phenomenological rather than mechanistic model. In molecular diffusion, the molecules move due to the kinetic energy of heat. What makes the cells/nuclei move here is not known. To know why they move in diffusive trajectories you would first need to know why they move at all. It is argued that they are not pushing each other around. For a model principally based on diffusion rather than packing or granular interaction, you need to posit that cells have a "heat" in the form of a natural random movement but given the density of the tissue, it is closer to a solid without diffusive movements of its constitutive molecules than a gas. Also, the decrease in diffusive movement when cell division is experimentally blocked argues against such a cell intrinsic heat. An extension to the main model, considers a "lattice" gas analogy but how this maps onto the tissue is not described.

Reviewer #3:

This is a review of the manuscript by Azizi et al., titled "Interkinetic nuclear migration in the zebrafish retina as a diffusive process". In this paper, the authors analyzed tracks of nuclear movements during early retinogenesis. These measurements were then used to describe retinal IKNM as a diffusive process across a nuclear concentration gradient. This manuscript is written in a very accessible way and was a pleasure to read. I only have a few comments.

1) To test predictions of the diffusive model, retinas were treated with aphidicolin and hydroxyurea, with the goal of preventing the nuclear diffusive flux at the apical boundary (inhibiting mitosis), and to prevent apical migration respectively. This abolished the concentration gradient (as predicted by the diffusion model). I wonder, however, if these experiments rule out a model of diffusion + basal drift, i.e. whether the treated drugs only inhibit what is assumed (mitosis and apical migration), or might also inhibit basal directed migration. I tried to understand the assumed mechanism of action of these drugs in this model system, but the Norden et al. paper which was cited to support the use of these drugs does not seem to have used them.

2) Norden et al. (2009) has already shown that the basal migration is a random walk, and that the apical migration is a persistent random walk (these two modes of migration have different signatures in the MSD versus time plots). Therefore, this paper seems a straightforward and logical extension from that work. Perhaps the authors could comment on how the measurements here improved upon the Norden et al. paper. Furthermore, if the measurements are significantly improved and/or different from the 2009 paper, it might be useful to replot MSDs versus time and demonstrate a lack of drift in the basal direction, and the presence of one in the apical direction similar to the Norden et al. paper.

3) While the authors allude to this point briefly in the Discussion, the model is not exactly accurate because it does not contain a convective term for the apical migration which opposes the direction of nuclear diffusion. This should be mentioned in the section which describes the model, and a clearer rationale provided for why this should not matter. I am not sure I completely understood the rationale for ignoring it, because even though the apical migration occupies only 8% of the cell cycle, physically that is the only mechanism for returning the nucleus to the apex for the next division to occur, and coupled with mitosis, is the basis for the increasing flux of nuclei at r = a with time.

[Editors’ note: further revisions were suggested prior to acceptance, as described below.]

Thank you for submitting your article "Nuclear crowding and nonlinear diffusion during interkinetic nuclear migration in the zebrafish retina" for consideration by *eLife*. Your article has been reviewed by Didier Stainier as the Senior Editor, a Reviewing Editor, and three reviewers. The following individuals involved in review of your submission have agreed to reveal their identity: Caren Norden (Reviewer #3).

The reviewers have discussed the reviews with one another and the Reviewing Editor has drafted this decision to help you prepare a revised submission.

Summary:

In the present manuscript the authors analyze interkinetic nuclear migration in the zebrafish retina. By long-term, rapid light-sheet and two-photon imaging during early retinogenesis they track entire populations of nuclei within this tissue. They show that crowding effects act in this process as nuclei reach close-packing and develop a nonlinear diffusional model that provides a quantitative account of the observations. Considerations of nuclear motion constrained inside the cell membrane are also used to show that concentration-dependent stochastic forces inside individual cells can offer a quantitative explanation of the nuclear movements observed during IKNM. By inhibiting cell cycle progression or changing temperature. They used their model to shed light on properties and mechanisms of the stochastic movements of nuclei during IKNM.

Essential revisions:

1) The revised manuscript puts much more emphasis on the "non-linear model", e.g. the title has been changed to include "non-linear". The authors should be clearer as to what is non-linear (MSD vs. time) and why this non-linearity matters. The conceptual intuition that nuclei are not points and face significant crowding at a certain density, should be well appreciated by readers.

2) The authors state they measured nuclear volume for 50 M-phase cells and saw no difference with other cell cycle phases, but did not include these data. The authors should both include this data and comment on it in the manuscript. This is an important, and not obvious, assumption in their model, and the literature shows some contradictory (though not authoritative) examples where nuclear volume does decrease from mother to daughters.

3) The authors should define apical crowding precisely. If for the authors, apical crowding equals a nuclear concentration gradient, do they mean to imply a mechanism like nucleokinesis, mitotic rounding, late G2 apical movement or simply an increase in density? The Introduction uses the term "apical crowding hypothesis" so this should be explicitly stated what this hypothesis is.

4) The authors lay big emphasis on the apical crowding hypothesis that leads to the basal-ward stochastic diffusive motion of nuclei. However, it has been shown in Matejcic et al., 2018 (Figure S3), that a maximum of 20% of the apical surface is covered by mitotic nuclei at any given timepoint, a ratio that would even be smaller at the early developmental stages the authors use in this study. Could it be discussed how this small apical occupancy and contribution to new material at the apical side fits with their diffusive model?

5) Along the same lines, while Figure 4 supposedly depicts the apical crowding effect, it would help to get an appreciation of this phenomenon also by original data that led to the graphs presented. Could the authors add images that show a difference between apical crowding between control and the HU-Aphidicolin condition qualitatively?

6) Along the same lines, Matejcic et al., 2018 showed that before 42hpf a basal exclusion zone exists that is not occupied by nuclei (due to an accumulation of basal actin). Did the authors take this exclusion zone into consideration for their analysis and in the model? How would this influence the model post 42 hpf, when nuclear occupancy spans the whole lengths of the apico-basal axis?

7) When describing and discussing their analysis the authors could make it clearer whether tracking was done in 2D or 3D. If I interpret Figure 1 and Figure 2 correctly, all tracking of nuclei was done in 2D using max projection or similar. This should be made clearer in the manuscript.

8) Could the authors explain better what they assume led to the difference between their data, that nuclei do not correlate speed of neighbors and the finding that blocking of apical nuclear migration slows all other nuclear movements as seen in Leung et al., 2011 in retinal tissue and Kosodo et al., in the neocortex.

9) The relation between IKNM and molecular motors and cytoskeletal elements the authors mention in the later part of their model is not very clear. Could they add some speculation on how they expect this to work? What cytoskeletal element and what type of motors are they referring to. The earlier study by Norden et al., 2009, that this study is building on, showed that no difference between velocity distributions or MSD exists for the stochastic part of motion during IKNM independently of whether microtubules are present or not. How does this fit with the authors interpretation that the stochastic motion is also driven by cytoskeletal elements? Also, it should be added how stochastic motor dependent transport could work in this scenario, as usually actin as well as microtubule dependent motors have a defined directionality.

---

## [Author Response]

[Editors’ note: the authors resubmitted a revised version of the paper for consideration. What follows is the authors’ response to the first round of review.]

Reviewer #1:The manuscript by Harris, Goldstein and colleagues use a combination of experiments and theory to describe retinal IKNM as a diffusive process. The movement of nuclei during G1 and S phase have previously been suggested to be passive and to be caused by apical crowding. The description of this process as a diffusive process is thus a logical extension of these previous proposals. That said, the strength of the present study is certainly more in providing a comprehensive physical description of the process rather than coming up with an entirely new concept.

The reviewer is indeed correct that previous work on IKNM had suggested that it was passive and caused by apical crowding. However, all previous (experimental) work examined only the trajectories of a few labelled nuclei in an otherwise unlabeled background. In this setting, it is not possible to investigate the effect of apical crowding, since crowding is an intrinsically collective effect that can only be studied when examining all of the nuclei in the tissue at once.

Furthermore, previous studies drew conclusions about the nature of the nuclear motion (i.e. whether it was active or passive) based solely on the behaviour of the mean-squared displacement (MSD) of nuclei during the individual phases of the cell cycle. However, the result that the MSD of a particle is quadratic in time for directed movements and linear in time for random movements only holds true in a “dilute” situation, where interactions between particles can be neglected, and under the condition that no boundaries are present with which the nuclei interact. Both conditions are clearly violated in the zebrafish retina. Therefore, we have to assume that the relationship between mode of nuclear motion and MSD is much more complex in IKNM than previously acknowledged. Our work provides an access point to studying the mode(s) of nuclear movement in IKNM by taking into account features of the IKNM and the retina previously neglected, namely

- the fact that a crucial feature of IKNM – the influence of crowding – is only possible to study by examining IKNM as a tissue-wide process;

- the 3D tissue shape, which is important because it provides much more space for the nuclei close to the apical side of the tissue than close to the basal side and therefore will influence the distribution of nuclei across the tissue;

- the nuclear interactions that are incorporated in the nonlinear version of our model;

- the fact that the number of nuclei and their packing density change over time. Because the nuclei are packed so tightly, the packing density most certainly influences the mobility and thus the motion of the nuclei. Moreover, because the packing density changes over time and across the apicobasal direction of the tissue, nuclear motion depends both on location within the tissue and developmental time.

The only major criticism I have is that the study is mainly descriptive with very limited functional interventions.

With all due respect, we find ourselves puzzled by this comment. “Descriptive” as a term in science usually refers to a mode of working that is the opposite of hypothesis-testing. Yet, our work is precisely focused on testing the hypothesis of nuclear crowding as a driver of IKNM, and it does that by combining carefully chosen experiments with development of a mathematical theory. In line with the recent essay in *eLife* by one of us (REG) on the role of theory in biological research (2018), we assert that such a mathematical development is an important result from which new physical and biological insight is gained, not just a form of description. We incorporate several features of IKNM and the retinal tissue that most certainly have an effect on the nuclear motion during IKNM and those effects have not been considered in previous work because of the experimental and mathematical limitations there. As a result, our work provides new insights into IKNM:

- Because previous studies have not studied IKNM as a collective phenomenon, it was not possible to distinguish properly between two possible modes of driving nuclear movements, namely motion created by apical moving nuclei that push other nuclei out of the way or motion created as a result of apical crowding. Based on our experimental work of tracking all the nuclei within a section of the retinal tissue and on careful analysis of these tracks, we were able to show that the first of the two suggested mechanisms is not operable in the zebrafish retina during IKNM. Our work supports the notion of apical crowding as a driving mechanism for IKNM.

- Although apical crowding had been suggested previously as a mechanism for driving IKNM (e.g. by Okamoto et al., 2013), the nuclear concentration, its gradient and thus the “crowding” have (to our knowledge) never explicitly been measured.

- Further, the proposed role that apical crowding has as a driver of IKNM had never been quantified before. In our work, we have developed a mathematical description of apical crowing and the distribution of nuclear concentration over time and across the retinal tissue. From this description, in contrast to previous work, we can estimate whether or not apical crowding alone is actually sufficient to create the nuclear distribution across the retinal tissue as observed in experiments. From the results of the model, we can conclude that apical crowding alone is indeed sufficient to explain the nuclear distribution across the tissue and over time.

- Although several studies have previously investigated IKNM as a random movement (at least during the majority of the cell cycle), these studies have neglected many important properties of the retinal tissue, as outlined before. Therefore, these studies might have come to conclusions that, in the light of our results, may need to be reconsidered. This specifically concerns any statements made based on mean-squared displacements (MSD) calculated from nuclear trajectories. These statements are fundamentally based on assumptions that are untenable in the zebrafish retina. Importantly, even the process of calculating a MSD-curve should be reconsidered. Because, as outlined above, the nuclear movements can be expected to depend on the local nuclear concentration and thus on space and time, one cannot simply merge nuclear trajectories from different time points or different locations across the retinal tissue into one MSD curve. The mathematical model outlined in our paper, in contrast, describes the process of IKNM in a manner consistent with these important properties and can be used in the future to analyze other aspects of IKNM.

- Several previous studies have investigated restrictions that might limit IKNM and thus possibly limit e.g. the cell cycle length in developing neuroepithelial tissues. For example, the accessibility of apical surface for nuclear divisions has been considered in Miyata et al., 2015. However, our work is the first, to our knowledge, that shows the importance of a maximum nuclear concentration on IKNM in the zebrafish retina.

Blocking S phase with HUA and changing temperature can have multiple side-effects that make the interpretations of the resulting phenotype more difficult. There have been more elegant genetic experiments in the past for interfering with apical crowding – for instance by knocking down TAG1 or overexpressing Wnt3 – that could have been used to challenge the predictions from the diffusion theory.

While the reviewer is correct about the possible genetic interventions to increase apical crowding, these are based on the genes expressed in other tissues (e.g. cortex) and species. To the best of our knowledge, there is no evidence of a role for TAG-1 (contactin-2) in the retina except in the context of axon guidance. Specifically, in mouse retinas, TAG-1 is expressed after RGCs have been born (e11.5). Furthermore, in cortical slices where TAG-1 has been used to create crowding, it does so by shorting the basal process leading to abnormal cell behaviour and detachment from the apical surface (Okamoto et al., 2013). Assuming TAG-1 is expressed earlier in zebrafish retinas and its knockout has a similar phenotype, shortening the basal process would presumably interfere much more severely with diffusion of nuclei given that they will not be able to move towards the basal surface at all. Regarding Wnt3, we cannot find any sources for its expression in the zebrafish retina. Wnt3 has been shown to promote proliferative divisions at the cost of neurogenic ones in other parts of the CNS. However, in the time window we are concerned with, all divisions of the retina are proliferative. Therefore, even if Wnt3 were expressed in the zebrafish retina, its overexpression would not be expected to increase overcrowding (unless it also shortens cell cycle length, for which we have not found any evidence).

The aim of the HUA experiment was not to increase overcrowding at the apical surface, but to stop it and see diffusion alone at work. This method has been used previously by a number of groups to interfere with IKNM (Murciano et al., 2002; Leung et al., 2011; Kosodo et al., 2011; Icha et al., 2016). It would be interesting to increase overcrowding and see if that would change the dynamics of the system. This is what we achieved with the temperature experiment were we increased and decreased crowding using a similar process, i.e. change in temperature. We agree with the reviewer that the change in temperature can have other side-effects. However, in addition to changing crowding at the apical surface, by considering changes in the diffusion properties of the nuclei at different temperature allows us to use our model to further probe possible molecular mechanisms for the observed stochastic diffusion.

Collectively, I think the study in its present form would in principle be suitable for publication although some more experimental work to address theoretical predictions would be preferable.

In light of the substantial additional theoretical results and data analysis now incorporated into the paper, we trust the referee will find it now suitable for publication.

Reviewer #2:In this manuscript the authors combine timelapse microscopy, cell tracking, and theory to develop a model of interkinetic nuclear migration in the pseudostratified epithelium of the developing zebrafish retina. It is argued that a theory based on diffusion in a concentration gradient can model the observed data.This paper fits in with several previous works which have used the zebrafish retina as a model for quantitatively looking at IKNM including ones that have measured the effective cellular diffusion constant (Leung et al., 2011) and argued that the movement is diffusive.My major concerns are with the assumptions of the diffusion model:1)What causes the gradient? In the model it is argued that there is a diffusion gradient because cells are added at the apical surface because that is where mitosis takes place.

To be clear, the argument is not about *cells* dividing at the apical surface, but about *nuclei* dividing at the apical surface.

However, when a cell divides, the daughters are typically half the volume of their mother and then double in size over the course of interphase. Daughter nuclei after division have half the DNA of their mother just before division so they are presumably smaller also, and would then grow during S-phase. It is important to measure nuclear volume as a function of cell cycle phase before asserting that apical mitosis is creating the gradient.

We have compared the volumes of 50 nuclei just before cell division with the volumes of 50 nuclei throughout the rest of the cell cycle and found no significant difference. This indicates that the nuclei do not simply halve their size during division and double it during S phase.

It seems apically directed movement of cells in late G2 could also create a concentration gradient although this is a small net movement that could be quickly relaxed.

There is good previous experimental evidence supporting the notion of the nuclear concentration gradient being created through apical crowding (e.g. Okamoto et al., 2013). However, neither had this gradient explicitly been measured before, nor had its effect on IKNM been investigated – namely creating differences in the nuclear concentration along the apical-to-basal direction of the tissue.

2) What causes the movement? The Abstract claims to "uncover the physical process" of IKNM which I don't think it does. It uses the same mathematical framework as would be used to study molecular diffusion as a physical process, but I would argue this is a phenomenological rather than mechanistic model. In molecular diffusion, the molecules move due to the kinetic energy of heat. What makes the cells/nuclei move here is not known. To know why they move in diffusive trajectories you would first need to know why they move at all. It is argued that they are not pushing each other around. For a model principally based on diffusion rather than packing or granular interaction, you need to posit that cells have a "heat" in the form of a natural random movement but given the density of the tissue, it is closer to a solid without diffusive movements of its constitutive molecules than a gas. Also, the decrease in diffusive movement when cell division is experimentally blocked argues against such a cell intrinsic heat. An extension to the main model, considers a "lattice" gas analogy but how this maps onto the tissue is not described.

Thank you for this question. We should clarify first that the fundamental starting point for the diffusive view of IKNM is that the individual trajectories of nuclear motion are noisy (‘stochastic’). This is not a postulate, but rather it is an observation. Whether the underlying cause of this stochasticity is thermal energy of the surrounding material (as in ordinary Brownian motion of microscopic particles in a fluid) or some more complex phenomenon associated with, say, the cytoskeleton acting on the nuclei, it is a general result in mathematical physics (given mild assumptions on the noise) that something like a diffusion equation emerges when one describes the concentration of such objects. This is why, for example, populations of bacteria executing run-and-tumble locomotion are also described by generalized diffusion equations. We should also clarify that in the simplest thermally-driven diffusion dynamics that it is the entropy of the spatial configuration that drives the diffusive spreading, and (as we now explain much more clearly in the revised manuscript) the lattice-gas model provides an estimate of the entropy when the nuclear packing is close to the its maximum, rather than being in the dilute limit associated with linear diffusion.

To be clear, the mechanistic elements of our model are:

- The effective influx of nuclei by nuclear division at the apical side of the tissue. If there were no such influx, the nuclear concentration gradient would simply diffusive away over time.

- The major influence of tissue architecture (as explained above).

- The major influence of steric nuclear interactions as incorporated via the maximum nuclear concentration c_max_ in the nonlinear model.

Reviewer #3:This is a review of the manuscript by Azizi et al., titled "Interkinetic nuclear migration in the zebrafish retina as a diffusive process". In this paper, the authors analyzed tracks of nuclear movements during early retinogenesis. These measurements were then used to describe retinal IKNM as a diffusive process across a nuclear concentration gradient. This manuscript is written in a very accessible way and was a pleasure to read. I only have a few comments.

Thank you for these comments.

1) To test predictions of the diffusive model, retinas were treated with aphidicolin and hydroxyurea, with the goal of preventing the nuclear diffusive flux at the apical boundary (inhibiting mitosis), and to prevent apical migration respectively. This abolished the concentration gradient (as predicted by the diffusion model). I wonder, however, if these experiments rule out a model of diffusion + basal drift, i.e. whether the treated drugs only inhibit what is assumed (mitosis and apical migration), or might also inhibit basal directed migration. I tried to understand the assumed mechanism of action of these drugs in this model system, but the Norden et al. paper which was cited to support the use of these drugs does not seem to have used them.

We apologize to the reviewer for having referenced the incorrect article. We have now corrected to this to reference Leung et al., 2011. We have now included a sentence explaining the mechanism of action of hydroxyurea and aphidicolin within the text. Since these drugs specifically arrest cell cycle at S phase by inhibiting DNA replication, their action should not directly interfere with any basal movement of nuclei.

2) Norden et al. (2009) has already shown that the basal migration is a random walk, and that the apical migration is a persistent random walk (these two modes of migration have different signatures in the MSD versus time plots). Therefore, this paper seems a straightforward and logical extension from that work. Perhaps the authors could comment on how the measurements here improved upon the Norden et al. paper. Furthermore, if the measurements are significantly improved and/or different from the 2009 paper, it might be useful to replot MSDs versus time and demonstrate a lack of drift in the basal direction, and the presence of one in the apical direction similar to the Norden et al. paper.

The fundamental difference from the 2009 work is that we track all the nuclei within a segment of the tissue, rather than studying the properties of a few labelled nuclei. This allows us to determine the nuclear concentration as a function of position and time, and thus to test the apical crowding hypothesis directly. Previous studies, including the work of Norden et al. (2009), have made statements about the nature of the nuclear motion, i.e. active or passive, based on assumptions that we believe do not hold in IKNM in the zebrafish retina. As we now explain in the revised Introduction and later in the paper, the relationships between the MSD and the underlying stochastic motion can be much more complex than suggested by the simplest linear diffusion problem associated with a non-interacting (‘dilute’) system without boundaries. Both assumptions appear to be violated in the zebrafish retina, and we have to assume that the relationship is much more complex in IKNM than previously acknowledged. Importantly, even the process of calculating a MSD-curve must be reconsidered, since, as outlined above, the nuclear movements depend on the local nuclear concentration and thus on space and time; one cannot simply merge nuclear trajectories from different time points or different locations across the retinal tissue into one MSD curve. The mathematical model outlined in our paper, in contrast, describes the process of IKNM in a manner consistent with the properties of dense nuclear packing and existing boundaries.

In the revised manuscript we have indeed re-analysed the MSD in light of the crowding hypothesis and the approach to close packing at late stages of IKNM. The new Figure 8 and discussion in the subsection “A stochastic model for the movement of individual nuclei reveals a potential microscopic mechanism for concentration-dependent IKNM”.

3) While the authors allude to this point briefly in the Discussion, the model is not exactly accurate because it does not contain a convective term for the apical migration which opposes the direction of nuclear diffusion. This should be mentioned in the section which describes the model, and a clearer rationale provided for why this should not matter. I am not sure I completely understood the rationale for ignoring it, because even though the apical migration occupies only 8% of the cell cycle, physically that is the only mechanism for returning the nucleus to the apex for the next division to occur, and coupled with mitosis, is the basis for the increasing flux of nuclei at r = a with time.

As the referee points out, apical migration involves only a very small part of the cell cycle and it could well be necessary to include an advective term in order to understand that particular phase in isolation. As we now make clear in the section “A diffusive model of IKNM”, our goal is to examine the simplest possible models for the evolution of the entire concentration profile to understand whether the apical crowding hypothesis is tenable, and that further work could indeed analyze the return motion separately along the same lines of the present study.

[Editors’ note: what follows is the authors’ response to the second round of review.]

Essential revisions:

1) The revised manuscript puts much more emphasis on the "non-linear model", e.g. the title has been changed to include "non-linear". The authors should be clearer as to what is non-linear (MSD vs. time) and why this non-linearity matters. The conceptual intuition that nuclei are not points and face significant crowding at a certain density, should be well appreciated by readers.

We thank the reviewer for this comment, but note that the terminology “nonlinear” is not about the MSD versus time. It is about the relationship between the flux of nuclei and the concentration gradient. We have added the following paragraph in subsection “Nonlinear extension to the model” to clarify the notion of nonlinearity in this context:

“The term "nonlinear" refers to the mathematical structure of the newly obtained Equation 9. In the mathematical classification, an equation is linear in a certain variable if this variable only appears to power 1 within the equation. For example, the simplest diffusion Equation 1 is linear in c and all its derivatives with respect to r and t_, like ∂c∕∂t._ In contrast, in Equation 9 the term c_max∕(_c_max−_c) appears which is proportional to c^−1^. Hence, Equation 9 is said to be nonlinear. The additional nonlinear term in Equation 9 (as compared to Equation 1) is an important aspect of the model as it arose from the introduction of the spatial extent of the nuclei and their maximum possible packing density c_max_. This effect also has to be taken into account in the boundary conditions.”

We have also added the phrases “(i.e. the spatial extent of nuclei in this case)” and “effectively introduce a particle size and, correspondingly, a maximum particle concentration” to subsection “Nonlinear extension to the model”, to further clarify the spatial nature of nuclei in the nonlinear model.

2) The authors state they measured nuclear volume for 50 M-phase cells and saw no difference with other cell cycle phases, but did not include these data. The authors should both include this data and comment on it in the manuscript. This is an important, and not obvious, assumption in their model, and the literature shows some contradictory (though not authoritative) examples where nuclear volume does decrease from mother to daughters.

We changed the text to add explicit measurements of nuclei before division, at two periods after division and at any random time during cell cycle starting in subsection “Analysis of nuclear tracks” to the following:

“We found no significant difference between average length of nuclear long axis when measured for 50 nuclei right before their division (5.0 ± 0.7 μm) compared to 50 others chosen randomly from any other time point within the cell cycle (5.3 ± 1.1 μm), indicating that this clearing is likely to have a biological explanation, such as the preferential occupancy of M phase nuclei at the apical surface during IKNM. We also performed the same measurements for 25 random nuclei 10 min after division when the average long axis length is significantly decreased by 0.8 fold (3.9 ± 0.5μm). However, this measurement increased significantly in the following 10 min (4.8±0.7μm) to become similar to that at M phase.”

In addition, we added the following sentences to the Discussion section as a further discussion of contribution of changes in nuclear size to establishment of the nuclear gradient: “While the newly divided daughter nuclei are approximately 0.8 fold smaller than M phase nuclei, they increase in size within 20 minutes of division to become statistically indistinguishable from M phase nuclei. Thus, the difference in the nuclear density apicobasally is unlikely to be a direct result of variability in nuclear sizes during cell cycle.”

3) The authors should define apical crowding precisely. If for the authors, apical crowding equals a nuclear concentration gradient, do they mean to imply a mechanism like nucleokinesis, mitotic rounding, late G2 apical movement or simply an increase in density? The Introduction uses the term "apical crowding hypothesis" so this should be explicitly stated what this hypothesis is.

We have added the following phrase to clarify the meaning of apical crowding to the Introduction: “i.e. an increase in nuclear packing density close to the apical tissue surface.”

4) The authors lay big emphasis on the apical crowding hypothesis that leads to the basal-ward stochastic diffusive motion of nuclei. However, it has been shown in Matejcic et al., 2018 (Figure S3), that a maximum of 20% of the apical surface is covered by mitotic nuclei at any given timepoint, a ratio that would even be smaller at the early developmental stages the authors use in this study. Could it be discussed how this small apical occupancy and contribution to new material at the apical side fits with their diffusive model?

The results presented in Matejcic et al., 2018 do not preclude the addition of nuclei at the apical surface, but rather comment on the rate at which this is done. Our conclusions are not based on the number of mitotic nuclei or on any specific rate of proliferation. Indeed, we consider retinal development at different temperatures where rates of proliferation are different and find our model to fit well for these conditions.

We have added the following sentences to the Discussion section to address this point: “Recent work indicates that only a small fraction of the apical tissue surface is occupied by mitotic cells at any given time, and thus retinal growth is not subject to a proliferative trap (Matejčić et al., 2018). Nonetheless, even this small fraction consistently adds to the number of cells at the apical surface (Figure 5A) contributing to the observed evolving gradient shown in Figure 6.”

5) Along the same lines, while Figure 4 supposedly depicts the apical crowding effect, it would help to get an appreciation of this phenomenon also by original data that led to the graphs presented. Could the authors add images that show a difference between apical crowding between control and the HU-Aphidicolin condition qualitatively?

We attempted adding the raw images to this figure; however, the differences are subtle and are not easily seen by eye in single frames, and 3D stacks are not helpful when illustrated in 2D. Therefore, we believe that the quantification presented is the best way of showing the change in nuclear concentration over time.

6) Along the same lines, Matejcic et al., 2018 showed that before 42hpf a basal exclusion zone exists that is not occupied by nuclei (due to an accumulation of basal actin). Did the authors take this exclusion zone into consideration for their analysis and in the model? How would this influence the model post 42 hpf, when nuclear occupancy spans the whole lengths of the apico-basal axis?

We do indeed account for the basal exclusion zone. The position where the basal boundary condition is applied is not set to the radius where cells actually have their basal surface but rather to the radius where nuclei appear to stop moving basally (i.e. at the basal edge of the most basal nucleus where the basal exclusion zone presumably begins). According to Matejcic et al., 2018, the basal exclusion zone only begins to vanish later in development than we consider here. We have also added the following paragraph to subsection “An analytical diffusion model of IKNM” to clarify this point to the readers:

“The position r = b where this basal boundary condition is applied could change throughout tissue development. Matejčić et al., (2018) found that a basal exclusion zone, where nuclei cannot enter due to accumulation of basal actin, exists in the zebrafish retina before approximately 42 hpf. Before this point in development, the no-flux boundary condition is applied at the tissue radius where the nuclear exclusion zone begins, while later in development, the no-flux boundary condition should be applied at the position of the actual basal cell surfaces. Since here we only model early stages of embryonic development well before the disappearance of the basal exclusion zone, the location r = b, where we apply our basal boundary condition, is chosen such that we only consider the region of the retinal tissue accessible to moving nuclei during IKNM.”

7) When describing and discussing their analysis the authors could make it clearer whether tracking was done in 2D or 3D. If I interpret Figure 1 and Figure 2 correctly, all tracking of nuclei was done in 2D using max projection or similar. This should be made clearer in the manuscript.

We thank the reviewers for pointing out this confusion. All tracking was carried out in 3D and we have added the phrase, “in 3 dimensions”, when mentioning the tracking pipeline in subsection “Generating image sets with high temporal resolution”. We also added “volume” instead of region in the caption reference to Figure 1C to clarify further this point.

8) Could the authors explain better what they assume led to the difference between their data, that nuclei do not correlate speed of neighbors and the finding that blocking of apical nuclear migration slows all other nuclear movements as seen in Leung et al., 2011 in retinal tissue and Kosodo et al., in the neocortex.

Blocking of nuclear migration removes the source of new nuclei at the apical surface by blocking G2 apical movement and mitosis. This would lead to a flattening of the concentration gradient (as in Figure 4A,B). The decrease in the slope of the concentration gradient would lead to a reduced collective diffusion, consistent with a decrease in the slope of the MSD graph, seen in Leung et al., 2011, as it measures the collective movement of nuclei. Therefore, our model is consistent with the observations in Leung et al. and Kosodo et al.

9) The relation between IKNM and molecular motors and cytoskeletal elements the authors mention in the later part of their model is not very clear. Could they add some speculation on how they expect this to work? What cytoskeletal element and what type of motors are they referring to. The earlier study by Norden et al., 2009, that this study is building on, showed that no difference between velocity distributions or MSD exists for the stochastic part of motion during IKNM independently of whether microtubules are present or not. How does this fit with the authors interpretation that the stochastic motion is also driven by cytoskeletal elements? Also, it should be added how stochastic motor dependent transport could work in this scenario, as usually actin as well as microtubule dependent motors have a defined directionality.

We have added the following paragraph to the Discussion section to discuss this issue further:

“Furthermore, disparate observations seem to agree with such an interpretation. For example, a microtubule cage was observed around RPC nuclei (Norden et al., 2009) and myosin was also shown to surround nuclei (Leung et al., 2011). Disruption of microtubule motor (dynactin-1) functionality either by mutation (Del Bene et al., 2008) or introduction of a dominant negative allele (Norden et al., 2009) leads to a more basal positioning of nuclei and occasional bursts of basal movement. A conjecture consistent with these observations would be that during G1 and S phases actomyosin based forces push the nucleus basally, as also seen in mouse telencephalon (Schenk et al., 2009), while microtubule motors push it apically, leading to a stochastic movement in both directions. Finally, in G2 concentration of myosin at the basal side of the nucleus leads to its rapid apical migration (Leung et al., 2011).”

We would like to point out that our microscopic interpretations based on forces involved in RPC IKNM is only a first step in exploring the contribution of cytoskeletal elements in the stochastic migration of nuclei. We acknowledge the need for further studies in this area immediately after the above paragraph.